# The impact of estimation methods for alcohol-attributable mortality on long-term trends for the general population and by educational level in Finland and Italy (Turin)

**Wanda Monika Johanna Van Hemelrijck**[1]*, **Pekka Martikainen**[2,3], **Nicolás Zengarini**[4], **Giuseppe Costa**[5], **Fanny Janssen**[1,6]

1 Netherlands Interdisciplinary Demographic Institute(NIDI)-KNAW/University of Groningen, The Hague, The Netherlands, 2 Population Research Unit, Faculty of Social Sciences, University of Helsinki, Helsinki, Finland, 3 Max Planck Institute for Demographic Research, Rostock, Germany, 4 Epidemiology Unit, ASL TO3, Piedmont Region, Grugliasco (TO), Italy, 5 Department of Public Health and Microbiology, University of Turin, Turin, Italy, 6 Population Research Centre, Faculty of Spatial Sciences, University of Groningen, Groningen, The Netherlands

* van.hemelrijck.wanda@gmail.com

**Data Availability Statement:** All data files containing the rates, ratios, and inequality indices described in the manuscript are available from the

## Abstract

### Background and aims

This paper assesses the impact of estimation methods for general and education-specific trends in alcohol-attributable mortality (AAM), and develops an alternative method that can be used when the data available for study is limited.

### Methods

We calculated yearly adult (30+) age-standardised and age-specific AAM rates by sex for the general population and by educational level (low, middle, high) in Finland and Turin (Italy) from 1972 to 2017. Furthermore the slope index of inequality and relative inequality index were computed by country and sex. We compared trends, levels, age distributions, and educational inequalities in AAM according to three existing estimation methods: (1) Underlying COD (UCOD), (2) Multiple COD (MCOD) method, and (3) the population attributable fractions (PAF)-method. An alternative method is developed based on the pros and cons of these methods and the outcomes of the comparison.

### Results

The UCOD and MCOD approaches revealed mainly increasing trends in AAM compared to the declining trends according to the PAF approach. These differences are more pronounced when examining AAM trends by educational groups, particularly for Finnish men. Until age 65, age patterns are similar for all methods, and levels nearly identical for MCOD and PAF in Finland. Our novel method assumes a similar trend and age pattern as observed in UCOD, but adjusts its level upwards so that it matches the level of the PAF approach for ages 30–64. Our new method yields levels in-between UCOD and PAF for Turin (Italy), and

**Funding:** This work is funded by the Netherlands Organisation for Scientific Research (NWO, https://www.nwo.nl/en) as part of the research project "Forecasting future socio-economic inequalities in longevity: the impact of lifestyle 'epidemics'", under grant no. VIC.191.019. See: www.futurelongevitybyeducation.com. PM was supported by the Academy of Finland (https://www.aka.fi/en/) (#308247, # 345219), the European Research Council (https://erc.europa.eu/homepage) under the European Union's Horizon 2020 research and innovation programme (grant agreement No 101019329), and the Max Planck – University of Helsinki Center for Social Inequalities in Population Health. The funders had no role in study design, data collection and analysis, decision to publish, or preparation of the manuscript.

**Competing interests:** The authors have declared that they have no competing interests.

resembles the MCOD rates in Finland for females. Relative inequalities deviate for the PAF-method (lower levels) compared to other methods, whereas absolute inequalities are generally lower for UCOD than all three methods that combine wholly and partly AAM.

## Conclusions

The choice of method to estimate AAM affects not only levels, but also general and education-specific trends and inequalities. Our newly developed method constitutes a better alternative for multiple-country studies by educational level than the currently used UCOD-method when the data available for study is limited to underlying causes of death.

## Introduction

Alcohol consumption strongly contributes to morbidity and mortality in many high income countries due to its effects on various acute and chronic causes of death [1–3]. Reliable information about alcohol-attributable mortality (AAM) is therefore crucial for policy makers to assess the harmful effects of alcohol in the population. Different estimation methods are available depending on the data sources at hand, yet no method to estimate AAM is considered the gold standard and, to date, no studies have compared how the methods impact findings on trends in AAM, whether for general populations or by educational attainment.

Methods can generally be categorised into those that include deaths from wholly alcohol-attributable conditions only, and those that include an estimation of partly alcohol-attributable mortality as well by considering those conditions for which at least some portion of deaths is likely to be due to alcohol consumption. The wholly underlying cause of death method (UCOD from here on) thereby only considers a selection of conditions which could only be caused by alcohol (e.g. accidental poisoning due to alcohol, alcoholic cardiomyopathy, alcohol dependence) [4, 5]. In contrast, the so-called multiple cause of death (MCOD) method considers the same list of causes, regardless of whether they appear as an underlying or a contributory cause of death on the death certificate. As such, the latter uses more elaborate cause-of-death data in order to include an estimate of partly alcohol-attributable conditions: those non-alcohol-specific conditions appearing as the underlying cause of death (e.g. road traffic accidents, self-harm, cardiovascular disease, and some respiratory conditions), that are nonetheless captured because the contributory cause is alcohol-specific [6, 7]. A third approach also captures an estimation of both wholly and partly alcohol-attributable mortality, by using population attributable fractions (PAFs) that are calculated from information on different levels alcohol consumption in a study setting (e.g. light, moderate, heavy drinking) as well as epidemiological information about relative risks of dying from a condition at those different levels of consumption. As such it determines which part of a large list of causes is attributable to alcohol, and thereby combines deaths accruing to health conditions which are wholly (equal to UCOD) and partially attributable to alcohol use (e.g. liver, breast and colorectal cancer, diabetes, stroke, pancreatitis, tuberculosis, unintentional injuries) [8].

Importantly, prior research has demonstrated important differences in levels and age patterns of AAM according to the method used, whereby methods relying only on wholly alcohol-attributable mortality (i.e. UCOD) resulted in rates only half of those approaches that attempt to incorporate partly-attributable mortality in different ways (i.e. MCOD, PAFs) [9]. Age patterns followed an inverse U-shape with peak age-specific mortality due to alcohol around age 65 for methods relying on information from death certificates only, and increasing

mortality with age for those including PAFs [9].How different methods influence trends over time and differentials by socio-economic status (SES) is, however, unknown. This assessment is nevertheless important, given that the method used likely affects our understanding about past developments and future developments in AAM for socio-economic subgroups.

Research on SES-specific AAM levels and trends, or socio-economic inequalities in AAM, has applied each of the abovementioned approaches to estimate AAM. To our knowledge, the only multiple-country European studies that included results on long-term trends in inequalities in AAM have relied on the UCOD-method [4, 10, 11], which by definition underestimates the total mortality burden due to alcohol consumption because it does not capture a wide range of causes of death that could be in part connected to alcohol consumption [9]. Single-country SES-specific studies from Finland and Spain have instead opted for MCOD methods [7, 12, 13]. Although this approach includes an estimate of partly alcohol-attributable mortality, contributory causes are not always available for study and the range of partly alcohol-attributable conditions included using this method may be limited at older ages in particular [6]. Alternatively, Italian SES-specific research has adopted a PAF-approach to estimate AAM by educational level [14]. PAF-approaches to estimate AAM are, however, also not void of limitations given that publicly available PAFs (e.g. from the Global Burden of Disease Study) are not education-specific and are considered of limited quality at older ages [9, 15]. Furthermore, the data needed to estimate PAFs that are both education- and country-specific (i.e. alcohol consumption by age, sex, and educational attainment; relative mortality risks at different levels of alcohol consumption) is rarely available in the country or region of study.

Despite these known differences between estimation approaches for AAM and their limitations, an assessment of differences in their resulting trends over time and by educational level is lacking. Given the substantial differences in levels and age-patterns previously demonstrated between various AAM estimation methods by Trias-Llimós and colleagues [9], and the potential policy implications of the AAM trends and educational patterns observed using any given method, this study assesses how (1) trends in AAM are affected by the choice of estimation method for the general population, as well as (2) the patterns by SES over time and the resulting inequalities. We do so for two European countries, namely Finland and Italy (Turin). In addition to addressing these two objectives, we (3) propose an alternative methodology to estimate AAM in multiple-country studies on AAM by SES when the data available for study is limited.

## Methods

### Data sources

We use individually linked cause-specific mortality information by educational level, sex, five-year age groups for single calendar years from Statistics Finland [16] and the Turin Longitudinal Study [17, 18]. For Finland, we dispose of underlying and contributory cause-of-death information to estimate AAM, whereby contributory causes have been registered since 1987. For Italy only underlying cause of death information is available for our entire study period. Turin will be referred to as Italy from here on. Prior evidence shows that the Turin data represent Italy well, also for alcohol-related mortality [19–21]. *S1 File*: *Data & Methods* provides a short overview of the data we used for each country in Table S1.1.

In both countries the study population is subdivided by SES according to individually collected census information on the highest level of completed education, whereby we consider International Standard Classification of Education 1997 (ISCED) 'Lower-' (ISCED 0–2; pre-primary to lower secondary education), 'Middle-' (ISCED 3–4; upper secondary to post-secondary non-tertiary education), and 'Higher-'educated (ISCED 5–6; tertiary education)

categories [22]. Although study evidence points to relationships between other measures of SES (e.g. income level, occupational status) and AAM in European countries [4, 23–25], levels and trends are very similar to those found for education [4]. Furthermore, we chose to define socio-economic groups based on education rather than occupational status or income given its wider availability over time and for women in particular, and its lower likelihood of reverse causation (i.e. worse health determining lower SES) [26, 27].

In addition to the individually linked education-specific cause-specific mortality data, we also obtained country-specific estimates of cause-specific alcohol-attributable fractions from the Global Burden of Disease (GBD) Study [28] by sex and five year age group (30–34, 35–39, 40–44, and so on, up to 95+) for single calendar years from 1990 up to 2017. These fractions were used to estimate AAM according to the PAF-method, and are not education-specific.

## Study population

We study AAM for the population aged 30 and older by educational level, sex, and age in Finland and Turin (Italy) for the period 1972 to 2017. These contexts represent two high-income countries from different European regions (i.e. Finland in the Nordic, Italy in the Mediterranean) that–within the European context and particularly from a historical perspective—portray vastly different past levels of alcohol consumption, overall drinking cultures, AAM levels, and SES inequalities within the continent, despite overall similar levels of current day per capita alcohol consumption [4, 29–33]. For example, levels have declined from 19.5 litres of per capita consumption of pure alcohol (age 15+) in Italy in 1972 to 7.4 litres in 2017, but have increased from 6.8 (1972) to 8.4 (2017) in Finland [33]. Furthermore In Italy, alcohol is more traditionally consumed with meals (mainly wine), while heavy episodic drinking (i.e. 8+ standard drinks per occasion among women, 10+ among men) is more common in Finland [34–36], and Finland generally still has a more permissive drinking culture than Italy does (e.g. perception of intoxication) [37]. Additionally, beer consumption has increased in both countries since the 1960s, while consumption of wine has declined in Italy in line with overall consumption declines, and consumption of spirits has become less common in both settings [37].

## Approach and existing estimation methods

We compared trends, levels, age distributions, and educational inequalities in AAM according to three existing estimation methods: the (1) Underlying cause of death (UCOD), (2) Multiple cause of death (MCOD) methods, and (3) the population attributable fractions(PAF)-method. We compare estimates of one method that only includes wholly alcohol-attributable mortality (UCOD), and two methods that include both wholly and partly alcohol-attributable mortality but measure partly alcohol-attributable mortality in different ways (MCOD and PAF).

The first method that solely relies on wholly alcohol attributable mortality that is purely cause of death (COD)-based and regards the following underlying causes of death with their corresponding International Classification of Disease 10[th] revision (ICD-10) as wholly alcohol-specific: mental and behavioural disorders due to alcohol use (F10, G31.2), alcoholic liver disease and cirrhosis (K70, K73, and K74), accidental poisoning by alcohol (X45), and alcoholic cardiomyopathy (I42.6). This *UCOD-method* with its list of 'alcohol-specific' causes of death was based on the list published by Rehm and colleagues [3]. However, we excluded causes of death that were not relevant to those aged 30 and older (e.g. P04.3 Foetus and newborn affected by maternal use of alcohol, Q86.0 Foetal alcohol syndrome), and excluded a few rarely occurring causes of death that caused issues for our study of trends over time due to ICD revisions. Compared to Mackenbach and colleagues [4], we added G31.2, K73 and K74 to account for ICD revisions and cross-country coding differences over time [38, 39]. We do not

expect these different selections of alcohol-specific conditions to majorly impact our findings, given that F10, K70, X45 and I42.6 together represent over 90% of the deaths considered entirely due to alcohol in Europe [5].

The second method includes wholly and partly AAM, whereby, the same list of causes as mentioned for the UCOD-method are considered wholly alcohol-attributable, regardless of whether they are listed as an underlying or as one of the three first contributory causes of death on the death certificate (the *MCOD-method*). Regardless of where on the death certificate the alcohol-specific cause appears, the death is included in its entirety. Contributory causes are listed by the coroner as being relevant to the process leading to death.

The *PAF method* also combines deaths accruing to health conditions which are wholly and partially attributable to alcohol use. It relies on alcohol attributable fractions from the GBD study [8], and multiplies death counts from an elaborate list of alcohol-related conditions that appear as an underlying cause of death on death certificates with their alcohol PAF. Importantly, the estimate for wholly AAM is the same as the UCOD method, whereby these conditions have a PAF of 1.00. Conditions that are partly alcohol-attributable, however, have a PAF<1.00. The country-, cause-, sex- and age-specific PAFs are calculated with information about alcohol-consumption levels and about cause-specific relative risks of dying at different levels of drinking versus abstainers. The GBD publishes negative PAFs for some causes of death by sex and age in Finland and Italy (i.e. ischaemic heart diseases for both sexes over age 60, stroke for women aged 65 and over). However, the potential for light to moderate alcohol consumption to prevent deaths is an area of academic debate [40] and we therefore replaced negative PAFs with 0.00 in the analyses of this paper. An overview of causes of death included in each method can be found in *S1 File*: *Data & Methods*, Table S1.2.

In order to compare age patterns and age-specific levels, we calculate age-specific rates clubbed by five-year periods as of 1990 (i.e. 1990–94, 1995–99, 2000–04, 2005–09, 2010–15) by sex and country for the general population and by educational level. To compare levels and trends, we furthermore calculated yearly age-standardised AAM rates by country and sex, also for the general population and by educational level. We perform direct age-standardisation using the in 2013 revised European Standard Population [41]. We furthermore calculate the relative inequality index (RII) and slope index of inequality (SII) for all methods to compare relative and absolute inequalities between them as well. For the RII, we adopted the Poisson regression-based approach by Moreno-Betancur et al. [42], and deduced the SII from the RII and age-standardised date rate in the total population [Formula 1] [12, 43].

$$SII_{y,c,s} = \frac{2 \times ASDR \times (RII - 1)}{(RII + 1)}$$ [1]

All outcomes were calculated for UCOD, MCOD, and PAF for Finland, and for UCOD and PAF for Italy. All calculations of mortality rates are done in Stata 17, whereas inequality indicators and figures are generated in RStudio 1.3.1093.

## Development of an alternative estimation method: The enhanced underlying cause of death method

In addition to the comparison of existing methods, we develop a method that is based on the advantages and disadvantages of the existing methods in the context of studying AAM by education or educational inequalities in AAM, and on the empirical comparison of their levels, age patterns, and trends.

The main advantages of the UCOD-method are its wide availability, given only underlying causes of death are required, as well as its education-specificity because the information used

for study comes directly from individual–and therefore education-specific–death certificates. However, the UCOD, by definition, considers only a narrow definition of AAM (i.e. only wholly AAM), and hence underestimates levels in AAM. The MCOD-method, in contrast, does provide an estimate of partially AAM that is also education-specific because it also relies solely on individual death certificates. Nevertheless, contributory causes of death are an additional piece of information that is not always available for study. The range of partly alcohol-attributable conditions that the MCOD includes for older age groups is furthermore limited. GBD PAFs used for the PAF-method are a publicly available alternative to including partly AAM in the estimates used for study, but alcohol attributable fractions are not education-specific and their overall quality beyond age 65 is questionable.

The 'enhanced underlying cause of death'-method (UCOD-Enh.) developed in this paper for multiple-country studies on AAM by SES with limited data relies on UCOD-data as a solid, widely available, and education-specific basis for AAM, while overcoming its underestimation of AAM by using a 'correction factor' that adds an estimation of partly AAM. This correction factor relies on PAF-data, but only for ages 30–64, and uses an average of the relationship between wholly and partly AAM between those ages over time to avoid any influence of time trends in non-education-specific fractions.

The development of the UCOD-Enh.-method is based on three main observations from the comparisons of existing methods (Figs 1–3). (a) Trends in AAM are virtually the same for the UCOD and MCOD in Finland (1987–2017); (b) age patterns for all methods are similar until age 65; (c) levels in AAM seem similar for the PAF and MCOD methods between ages 30–64, and proportionally higher compared to UCOD AAM rates.

Our new method uses UCOD rates as the baseline for the age pattern and trend in AAM, but its levels are adjusted upwards to resemble PAF levels between ages 30 and 64 and thereby include an estimate for partly AAM. This resemblance is deemed reasonable to strive for given (a) the levels for both methods that include wholly and partly AAM (MCOD and PAF) were similar between ages 30 and 64 across educational levels, (b) because PAF-estimates at higher ages are less optimal and are not based on education-specific information regardless of age, and (c) contributory cause of death data is not available for many countries (e.g. Italy for our study period).

The level-adjustment consists of a ratio-based approach, whereby the ratio summarises the relationship between UCOD (wholly AAM) and PAF (wholly and partly AAM) between 1990 and 2017 (age-standardised death rates [ASD] truncated for ages 30–64) [Formula 2]. The average ratio thereby amounts to 1.98 among Finnish males, 1.86 for Finnish females, and 3.43 and 3.33 for Italian males and females, respectively. The UCOD death counts (D_UCOD) by country, year, educational group, sex and age are multiplied with a time-constant sex- and country-specific ratio of age-standardised PAF rates over UCOD rates between ages 30–64 [Formula 3].

$$Ratio_{c,s} = \sum_y \left( \frac{ASDR3064_{PAF\,country,sex}}{ASDR3064_{UCOD\,country,sex}} \right) / N_y \qquad [2]$$

$$D\_UCOD\_Enh_{c,s,e,a} = D\_UCOD_{c,s,e,a} \times Ratio_{c,s} \qquad [3]$$

The development of the method is described in more detail in S2 File: Enhanced underlying cause of death method, where an overview of ratios can also be found (Table S2.1). We demonstrate how levels, age patterns, trends and inequalities for our novel method compare to those of the previously described estimation methods.

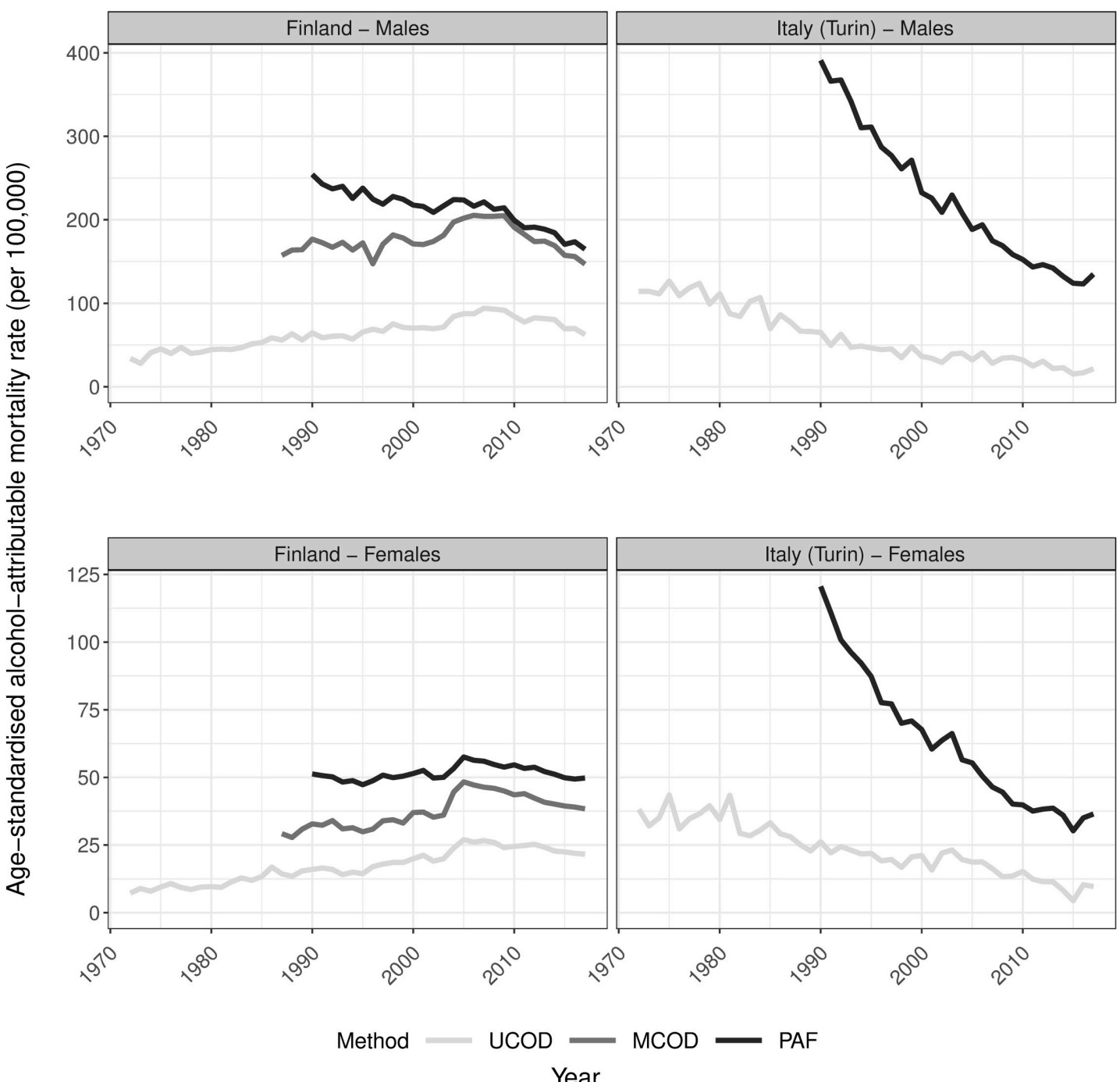

**Fig 1. Trends in age-standardised alcohol-attributable mortality rates by country and sex for ages 30 and older according to different estimation methods, 1972–2017.** UCOD = 'Underlying cause of death', MCOD = 'Multiple cause of death', PAF = 'Population-attributable fraction-based'; Rates expressed per 100,000 person years; The Y-axis scale differs by sex to improve visibility of the results; Data sources: Statistics Finland, Turin Longitudinal Study.

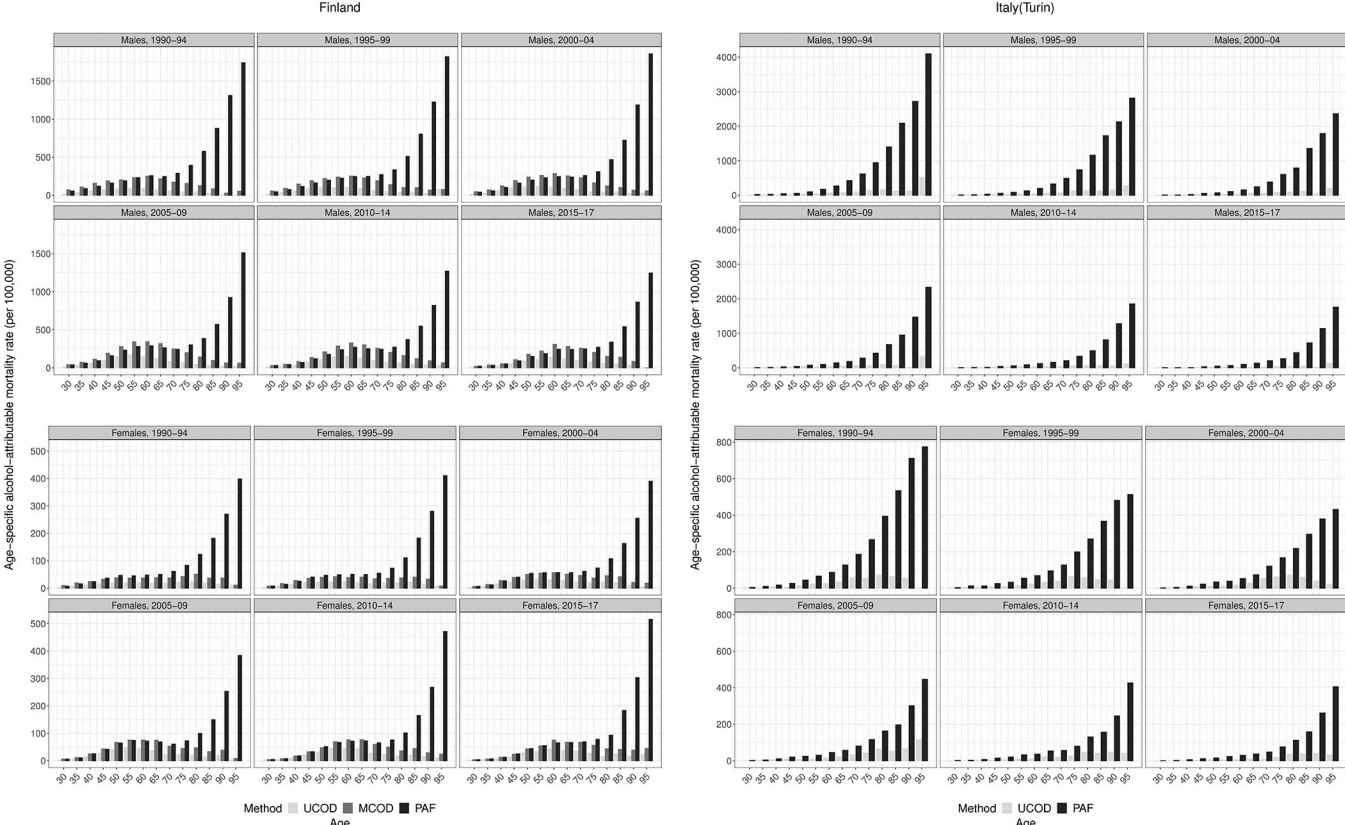

**Fig 2. a. Age patterns in alcohol-attributable mortality by country and sex for ages 30 and older according to different estimation methods, clubbed rates by five year period (excl. 2016–17), 1990–2017, Finland.** UCOD = 'Underlying cause of death', MCOD = 'Multiple cause of death', PAF = 'Population-attributable fraction-based'; Rates expressed per 100,000 person years; The Y-axis scale differs by sex to improve visibility of the results; ages 30 and older: 30–34, 35–39, 40–44, and so on, up to 95+; Data sources: Statistics Finland. **b. Age patterns in alcohol-attributable mortality by country and sex for ages 30 and older according to different estimation methods, clubbed rates by five year period (excl. 2016–17), 1990–2017, Turin(Italy).** UCOD = 'Underlying cause of death', PAF = 'Population-attributable fraction-based'; Rates expressed per 100,000 person years; The Y-axis scale differs by sex and country to improve visibility of the results; ages 30 and older: 30–34, 35–39, 40–44, and so on, up to 95+; Data sources: Turin Longitudinal Study.

## Results

### The impact of estimation methods for alcohol-attributable mortality on findings in the general population

Our results confirm the effect of the estimation method on AAM levels and age patterns. The UCOD reveals far lower levels of AAM compared to the MCOD and the PAF methods (Fig 1), for which levels are more alike (especially for Finnish males). Regarding the age pattern (Fig 2A and 2B for the general population, Figs S3.1a and S3.1b by educational level in *S3 File: additional figures*), the COD-based methods in Finland reveal a similar increase with age followed by a decline after about 65 years, whereas the PAF method is similar up until that, but increases exponentially after that. Age-specific levels in AAM are virtually the same for PAF and MCOD between ages 30–64 in Finland, whereby they appear proportionally higher (on a log scale) than UCOD levels. The lower levels of UCOD and exponential increase in AAM with age according to the PAF method is also discernible in the Italian age pattern of AAM for both sexes.

Regarding trends over time, the COD approaches show similar–generally less favourable (i.e. steeper increases and slower declines)—trends in AAM compared to the PAF method

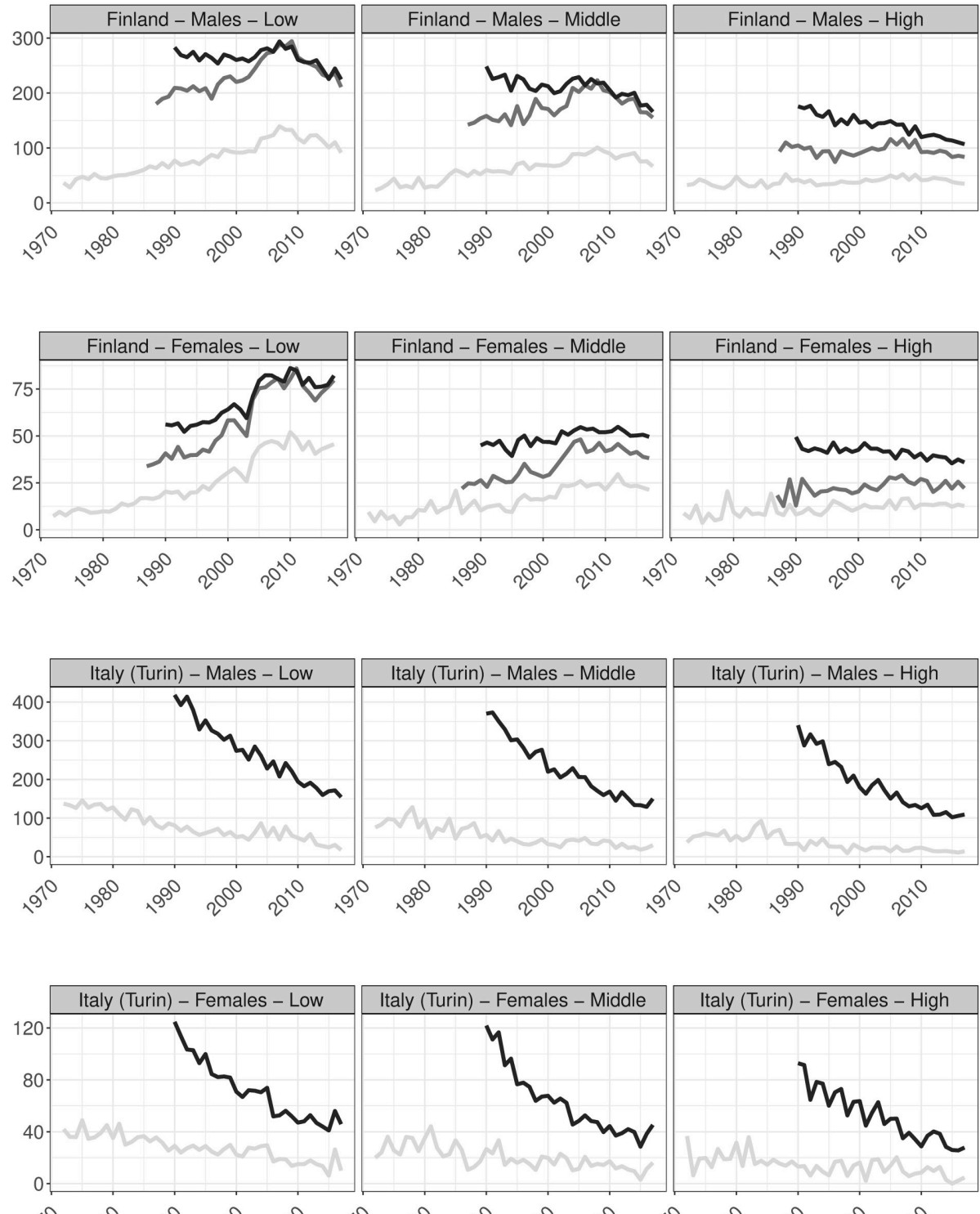

**Fig 3. Trends in age-standardised alcohol-attributable mortality rates by country, sex, and educational level for ages 30 and older according to different estimation methods, 1972–2017.** UCOD = 'Underlying cause of death', MCOD = 'Multiple cause of death', PAF = 'Population-attributable fraction-based'; Rates expressed per 100,000 person years; Y-axis scale differs by sex and country to improve visibility of the results; Data sources: Statistics Finland, Turin Longitudinal Study.

(Fig 1). For Finland we observe continuous increases up to about 2005–2008 with declines thereafter for the two COD approaches, whereas the PAF method declines among males and is more stable among females throughout. For Italy, the overall declines we find in AAM are less pronounced for UCOD compared to PAF trends.

## The impact of estimation methods for alcohol-attributable mortality on findings by educational level and on educational inequalities

The overall less favourable trends in AAM according to COD approaches are more visible when trends are studied by educational group (Fig 3). For both sexes in Finland, for example, we observe the same trends we found for the general population among the lower-educated, but differentials are more pronounced. For the middle- and higher-educated, however, the PAF method demonstrates clear declines in AAM that we do not find using COD methods.

For Italy, PAF trends reveal declines in AAM in all educational groups, whereby the educational trends somewhat converge throughout the study period. For the UCOD method, AAM also declines regardless of educational attainment, but the pace of decline appears to differ more b educational level. Overall, both approaches imply educational convergence of AAM trends in Italy, with small absolute mortality gaps in recent years (see also additional Fig S3.2 in S3 File). Given the larger levels of AAM using PAF than UCOD early on in the study period, however, declines in absolute mortality inequalities are likely more pronounced using the former than the latter method.

## An alternative estimation method for alcohol-attributable mortality

Fig 4 compares the levels and trends in age-standardised AAM (30+) for our new UCOD-Enh. method to the existing methods in the general population and by educational level. For the general population, trends are–by design–identical to the those in UCOD, but at higher levels that are generally below the levels of the PAF approach. In Finland, the trends and levels for the new method are nearly identical to the MCOD for females, but levels are slightly lower among males. This is also visible at the age-specific level (additional Fig S3.1a in S3 File), where rates are consistently lower for UCOD-Enh. than for MCOD among Finnish males. In Italy, the age-specific levels are similar to those using PAF between ages 30 and 64, but lower at older ages (additional Fig S3.1b in S3 File), which results in lower age-standardised levels and less steep trends in AAM.

By educational level, the new method demonstrates a steeper increase followed by a decline after 2008 for lower- and middle-educated males in Finland than the UCOD and MCOD methods do, while the PAF-method shows declines throughout. Among Finnish females, developments by educational level are similar for the new method compared to UCOD and MCOD. PAF-trends are, however, different: they increase less steeply among the low-educated, are overall higher and more stable for the middle-educated, and decline among the high-educated in contrast to the other three methods. In Italy, the new method still shows declining AAM in all educational groups, but does so at levels in-between UCOD and PAF.

Concerning inequalities, Figs 5 and 6 reveal increasing relative (RII) and absolute (SII) inequalities in UCOD-Enh. based AAM for both sexes in Finland until the mid-2000s, after which they stagnate (RII) or even decline (SII). Levels in relative inequalities are thereby

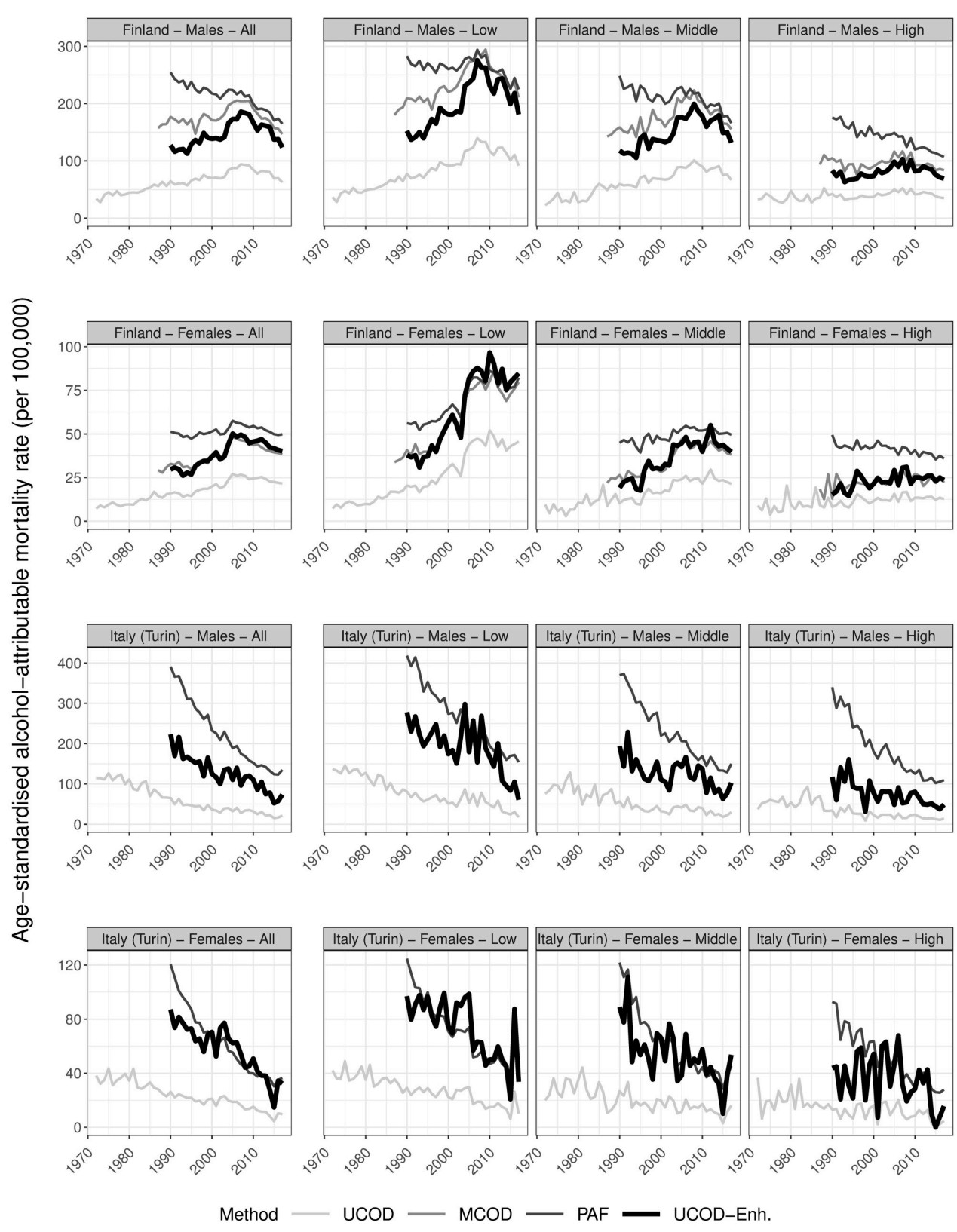

**Fig 4. Trends in age-standardised alcohol-attributable mortality rates by country, sex, and educational level for ages 30 and older according to different estimation methods, 1972–2017, including the newly developed 'Enhanced UCOD' method.** UCOD = 'Underlying cause of death', MCOD = 'Multiple cause of death', PAF = 'Population-attributable fraction-based', UCOD-Enh. = 'Enhanced UCOD'; Rates expressed per 100,000 person years; The Y-axis scale differs by sex to improve visibility of the results; Data sources: Statistics Finland, Turin Longitudinal Study.

similar between UCOD, MCOD, and UCOD-Enh and somewhat lower for PAF. The levels of absolute inequalities are similar for MCOD, PAF, and UCOD-Enh., but lower for UCOD.

In Italy, inequalities in UCOD-Enh. strongly fluctuate over time for both sexes, and both absolute and relative outcomes (observed dots). However, absolute inequalities generally decline, and judging from smoothed trends these declines are most pronounced using the UCOD-Enh. method (Fig 6). Similar to Finland, relative inequalities are generally lower using PAF than other methods for both sexes.

## Discussion

### Summary of results

The findings of this study revealed that, first, visual analysis of trends yields highly similar curves when only COD data is used, both in the general population and by educational level, albeit at different levels. UCOD and MCOD thereby showed mainly increasing trends in AAM according to UCOD and MCOD compared to a generally declining trend according to the PAF-method. Second, age patterns are similar until age 65 for all methods, albeit at different level. Third, levels in AAM are higher for any method that includes an estimate of both wholly and partly AAM (MCOD and PAF) than wholly AAM only (UCOD), with a seemingly proportionally higher level for PAF and MCOD between ages 30–64 compared to the UCOD method. These observations regarding trends, levels, and age patterns for different methods are even more pronounced when trends in AAM are examined by educational level. Our novel method that uses UCOD deaths as a basis, but employs a multiplier to increase its levels upwards as a proxy for partially alcohol-attributable mortality, follows age patterns and trends of the solely COD-based methods (UCOD and MCOD), but approaches levels of methods that combine wholly and partly AAM (MCOD and PAF). Finally, the PAF method stands out as resulting in lower relative inequalities than the other three methods (similar to its different trends in AAM), whereas absolute inequalities are higher using a method that combines wholly and partly AAM (MCOD, PAF, UCOD-Enh.).

### Interpretation of our findings

Our study results show that the method to estimate AAM does not only affect levels and age patterns, but also trends over time, both for the total population, and for education groups.

Our finding that AAM *trends* are generally more favourable using a PAF-based approach compared to a COD-approach (due to general declines in Finland for males and steeper declines in Italy than using another method) is most likely due to its reliance on different data (i.e. PAFs), and particularly due to the inclusion of a range of cardiovascular diseases (CVDs) (e.g. ischaemic heart disease, stroke, hypertension) in PAF methods that are not typically included in COD-based methods, but carry a strong weight in our PAF estimates. Indeed, CVDs other than alcoholic cardiomyopathy are not included in UCOD estimates and only included in MCOD when alcohol is considered a contributory cause of death [6, 44]. Given CVD mortality has declined rapidly since the 1970s [45], trends in PAF rates in particular also decline (*S3 File: Additional figures*, Fig S3.3c). We therefore suggest that the declining AAM mortality shown by the PAF method among Finnish males in particular does not solely reflect the role of changes in alcohol consumption. The more pronounced declines in CVD mortality

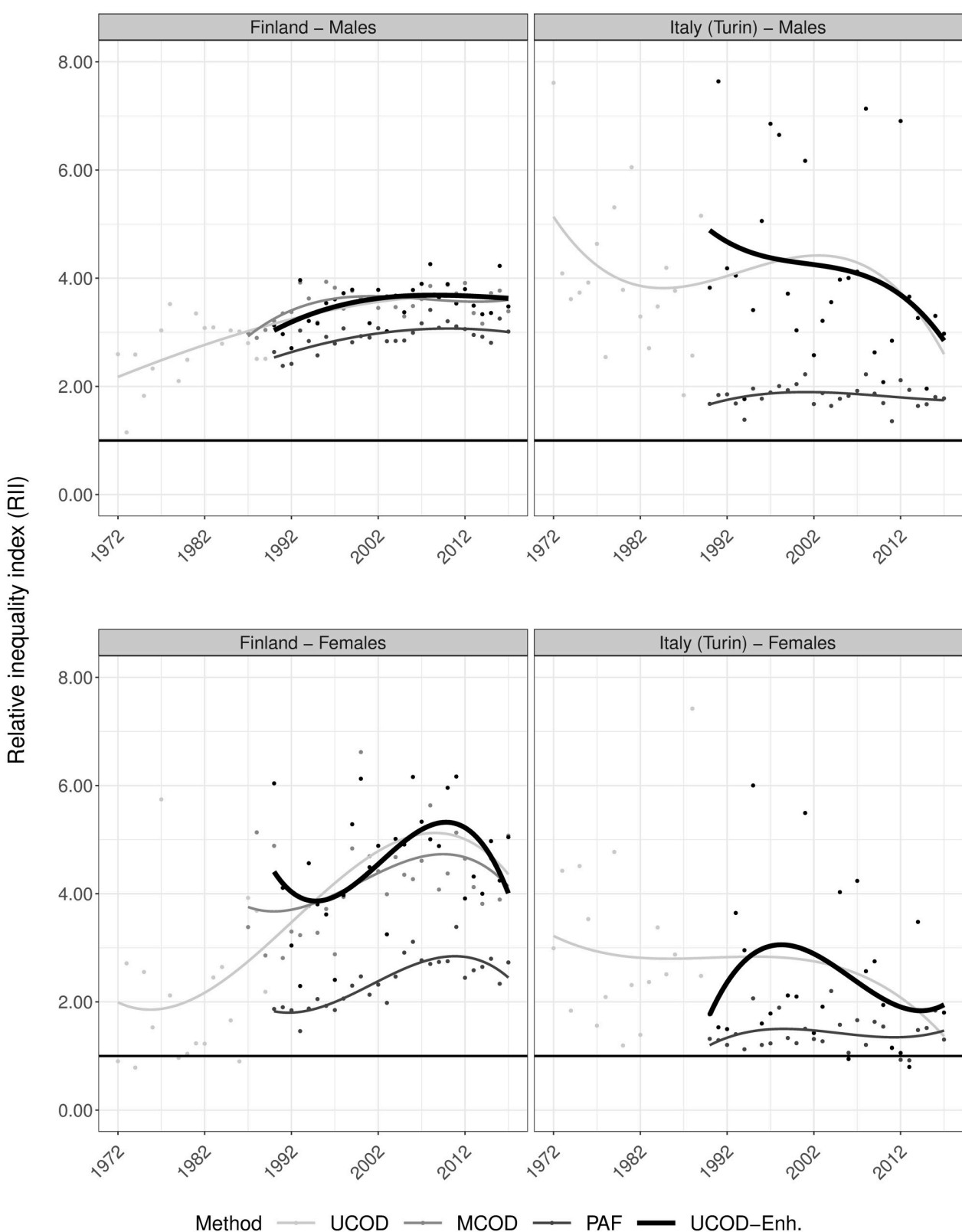

**Fig 5. Trends in relative educational inequalities (relative inequality index) in alcohol-attributable mortality by country and sex for ages 30 and older according to different estimation methods, including a newly developed one, 1972–2017.** Dots represent observed values, lines are smoothed trends using cubic splines. UCOD = 'Underlying cause of death', MCOD = 'Multiple cause of death', PAF = 'Population attributable fractions-based', UCOD-Enh. = 'Enhanced underlying cause of death'; SII expressed per 100,000 person years; Outliers (RII>8) among Italian females for the UCOD and UCOD-Enh. methods in 1987, 1998, 2004, 2015, and 2016 were excluded to enhance visibility of the overall figure; Data sources: Statistics Finland, Turin Longitudinal Study.

among higher versus lower SES groups (e.g. according to occupational status) observed in prior research for Finland during the cardiovascular revolution [46] also seem observable in our education-specific trends, whereby declines in PAF are particularly pronounced for the middle- and higher-educated. Importantly, the trends by educational attainment we demonstrated for the PAF-method are generally less trustworthy. The data required to come by a completely country- and education-specific PAF-estimate are difficult to come by, namely alcohol consumption data that is specific to educational attainment, and education- and country-specific relative mortality risks by different levels of alcohol consumption compared to abstainers. Future research relying on PAF-methods to estimate AAM by educational level would benefit from overall improvements in the specificity of PAFs by age group and country (particularly in the relative mortality risks used) in order to avoid bias in the role attributed to alcohol in mortality trends over time, but more crucially also from the development of education-specific fractions.

The AAM methods used also affect conclusions drawn regarding educational inequalities in AAM. Our results demonstrated comparable relative inequalities between UCOD, MCOD, and UCOD-Enh. methods, but lower ones for PAF, again pointing to a different result when non-education-specific information is used Crucially, especially absolute inequalities will be sensitive to the degree to which a method captures only wholly or also an estimation of partly alcohol-attributable mortality, whereby inequalities were higher for MCOD, PAF, and UCOD-Enh. methods than for UCOD in both settings and for both sexes. This implies that relative inequalities are likely to be comparable for methods using education-specific information throughout but not for the PAF method that does not use education-specific RRs and alcohol consumption, but that absolute inequalities measured by SII using UCOD as demonstrated by Mackenbach, Kulhanova [4] will be underestimated due to a narrow definition of AAM.

## Appraisal of the new method

Our newly developed UCOD-Enh. AAM estimation method was based on known benefits and limitations of existing methods, and our detailed comparison of levels, age patterns, and trends in those methods. This led us to use UCOD as the most likely data available as basis for our new method, whereby it assumes a similar trend and age pattern as observed in UCOD, but adjusts its level upwards so that it matches the level of the PAF approach for ages 30–64 and includes an estimate of both wholly and partly AAM.

Our method hence generally provides a better alternative to the UCOD-method given its narrow definition of AAM, and to the PAF-method given known issues with the calculation of PAFs due to alcohol at older ages. Particularly in studies aiming to study socio-economic differences and inequalities in AAM, the method relies less on non-education-specific PAFs than the PAF-method does. The main principles used should nevertheless not remain unmentioned in this manuscript.

First, our new method considers the trend and age pattern in UCOD are reliable, but levels underestimated given its narrow definition of AAM. Indeed, our and prior research results revealed highly similar trends and age patterns in solely COD-based methods (Figs 1 and 2A), and reported that age patterns in UCOD are more reliable than others because the data used

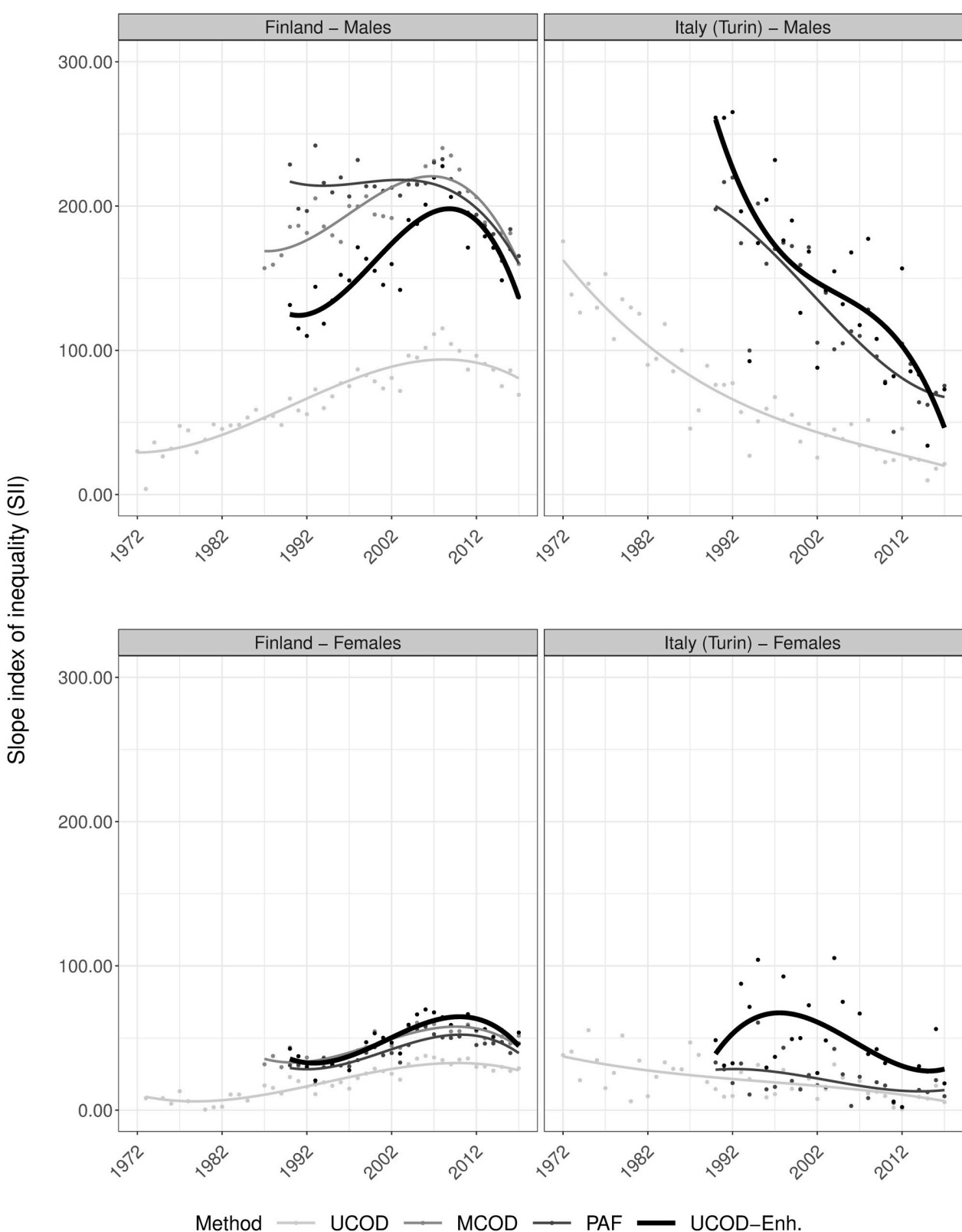

**Fig 6. Trends in absolute educational inequalities (slope index of inequality) in alcohol-attributable mortality by country and sex for ages 30 and older according to different estimation methods, including a newly developed one, 1972–2017.** Dots represent observed values, lines are smoothed trends using cubic splines. UCOD = 'Underlying cause of death', MCOD = 'Multiple cause of death', PAF = 'Population attributable fractions-based', UCOD-Enh. = 'Enhanced underlying cause of death'; SII expressed per 100,000 person years; Data sources: Statistics Finland, Turin Longitudinal Study.

does not rely on any additional assumptions about AAM, and country differences in coding will generally be limited to liver disease mortality [9, 47]. Although there are clear differences in the UCOD age patterns between Finland and Italy, whereby AAM declines abruptly with age in Finland but not in Italy, these differences can be explained by different patterns of drinking, with alcohol consumption at large being higher in Italy and cancers playing a larger role at older ages, but the drinking behaviours being riskier and occurring at younger ages in Finland [37]. Of the existing methods, we furthermore assumed the trends in UCOD most reliable given underlying causes of death are generally registered on death certificates, and subject to less changes in practice over time and between countries than contributory causes of death are for the MCOD method, or the survey data needed to calculate alcohol attributable fractions for the PAF method.

Second, we assumed that the PAF-based AAM levels for ages 30–64 are reasonable estimates of wholly and partly alcohol-attributable mortality at those ages, but not at older ages given research reporting on unreliable PAF-estimates above age 64 [9, 15, 29]. We therefore chose to avoid any reliance on PAF-based AAM after age 65 when developing a multiplier for AAM levels for our new method.

Third, our method assumes a proportional relationship between levels of fully and partly alcohol-attributable deaths combined, and fully alcohol-attributable deaths alone. We based this principle on the level difference we observed between age-standardised UCOD- and MCOD-rates, as well as age-specific UCOD- and MCOD on the one hand, or PAF-rates until age 64 on the other hand. This difference was consistently observed as a relative one, whereby multiplying the UCOD rates with a same ratio in each year would theoretically result in the same levels as MCOD rates at the age-specific or -standardised levels, or PAF at age-specific levels between ages 30 and 64. We find this relationship particularly visible for Finnish females in Figs 1 and 2A, and in line with Trias-Llimós and colleagues [9] demonstrating a comparable proportionality in Finland and France for both sexes in 2013. Importantly, this is the easiest to observe and apply relationship between wholly and partly alcohol-attributable mortality, and therefore serves as the main justification or our ratio-based approach in settings where limited data is available (e.g. only UCOD and PAFs from the GBD Study). Nevertheless, this proportionality is may not be a universal observation, which can also be an important limitation of UCOD-Enh. in some settings. For example, the long-term trend in AAM for Finnish males reveals deviations from this proportionality over time (Figs 1, 2A and 3). It is likely that the addition of external causes using MCOD [6, 44], especially among lower educated men, was larger prior to the 2000s than a proportional increase using our UCOD-Enh. could capture because external causes were only partially included in PAF (which UCOD-Enh. partly relies on) but fully counted in MCOD (*S3 File*, Fig S3.3b). We recommend a careful assessment of this proportionality principle using the data available in other study settings prior to implementing this ratio-based approach.

Importantly, our newly proposed UCOD-Enh. method applies the abovementioned assumptions in the same way for all educational groups. It does so because education-specific alcohol attributable fractions have not been developed (longitudinally), and would require high-quality survey data on alcohol consumption and relative mortality risks for different levels of consumption by educational level over time and by sex and age in various countries.

UCOD-Enh. principle are therefore not education-specific, although ideally they would be. Given the reliance of this method the quality of the available attributable fractions, we therefore still recommend using AAM estimation methods that rely on more extensive information when it is available, especially information that is education-specific (e.g. contributory causes of death). When more elaborate data is lacking for long periods of time by educational level, however, the UCOD-Enh. method can serve as an alternative because it uses a less narrow definition of AAM than UCOD does, and avoids reliance on suboptimal alcohol attributable fractions at older ages in PAF. Although an in-depth interpretation of the UCOD-Enh. trends in terms of how different causes of death contribute to AAM over time is not feasible, the method relies on as much high-quality information on trends, age patterns, and levels in AAM as possible while remaining easy to apply.

## Limitations

Although our study includes two European countries only, choosing a small selection has allowed us to perform an in-depth comparison of estimation methods for multiple outcomes (i.e. levels, age patterns, trends, for the general population and by educational level), and our choice of countries includes–within Europe and from a historical perspective—two very different cultures when it comes to alcohol consumption, with different consumption trends and levels.

We cannot, however, exclude that potential changes in coding practices over time and differences therein between countries may have affected our results, in addition to the possibility of (changes in) socio-economic bias in cause-of-death coding related to alcohol. Particularly differing levels of training and stigma surrounding alcohol dependence (as captured by ICD-10 code F10) occur between countries [48, 49], and arguably changed throughout our study period. Additionally, Mäkelä and colleagues [50] have suggested that socio-economic bias may be present in diagnostic practices of medical doctors when it comes to alcohol-related issues, whereby they would be less inclined to diagnose someone of similar SES to their own with alcohol dependence, which may extend to defining the cause of death on a death certificate as alcohol-related. As a result, we chose not to compare trends and levels in AAM (by educational level) between Finland and Italy, but rather focus on the in-country comparison between methods. Although beyond the scope of this study, an in-depth analysis of potential effects of socio-economic bias on trends in AAM by education in different countries would help unveil potential and lesser-known data artefacts. Furthermore, as new PAFs become available in the academic literature, estimations relying on PAF will also change. Future research comparing different methods will thus be needed in the years to come.

## Overall conclusion

The technique used to estimate alcohol-attributable mortality affects not only levels and age patterns, but also trends over time, both for the general population and by socio-economic group. Consequently the method used also affects conclusions drawn regarding socio-economic inequalities in mortality. There being no gold standard to estimating alcohol-attributable mortality, our results furthermore dissuade reliance on the underlying cause of death method for studies on SES-specific trends in alcohol-attributable mortality, as well as on non-educations-specific PAFs that are furthermore limited at older ages. Although not without limitation, the multiple cause of death method may provide a good option for this type or research, and the newly developed method in this paper may provide another alternative for multiple-country studies by SES where no elaborate cause of death information or education-specific PAFs are available.

## Supporting information

**S1 File. Data and methods.**
(DOCX)

**S2 File. Enhanced underlying cause of death method.**
(DOCX)

**S3 File. Additional figures.**
(DOCX)

## Acknowledgments

The authors thank David Anthonio (BSc in Econometrics and Operations Research, University of Groningen; Netherlands Interdisciplinary Demographic Institute (NIDI)) for his assistance in making figures, and Daniel Zazueta (NIDI) for providing his R scripts to perform the Rizzi et al. smoothing. The authors alone are responsible for the interpretation of the data.

## Author Contributions

**Conceptualization:** Wanda Monika Johanna Van Hemelrijck, Fanny Janssen.

**Data curation:** Wanda Monika Johanna Van Hemelrijck, Fanny Janssen.

**Formal analysis:** Wanda Monika Johanna Van Hemelrijck.

**Funding acquisition:** Fanny Janssen.

**Methodology:** Wanda Monika Johanna Van Hemelrijck, Pekka Martikainen, Fanny Janssen.

**Project administration:** Wanda Monika Johanna Van Hemelrijck, Fanny Janssen.

**Software:** Wanda Monika Johanna Van Hemelrijck.

**Supervision:** Fanny Janssen.

**Validation:** Wanda Monika Johanna Van Hemelrijck, Fanny Janssen.

**Writing – original draft:** Wanda Monika Johanna Van Hemelrijck, Fanny Janssen.

**Writing – review & editing:** Pekka Martikainen, Nicolás Zengarini, Giuseppe Costa.

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
