## [Decision Letter · Decision Letter 0]

2 Aug 2023

PONE-D-23-15590The impact of estimation methods for alcohol-attributable mortality on long-term trends for the general population and by educational level in Finland and Italy (Turin)PLOS ONE

Dear Dr. Van Hemelrijck,

Thank you for submitting your manuscript to PLOS ONE. After careful consideration, we feel that it has merit but does not fully meet PLOS ONE’s publication criteria as it currently stands. Therefore, we invite you to submit a revised version of the manuscript that addresses the points raised during the review process.

The paper is of great value, but recommendations for improvements have been made by reviewers. After consideration of their comments, we find it necessary to request major revisions to ensure the manuscript meets the required standards for publication.

We look forward to receiving your revised manuscript.

Kind regards,

Elisenda Renteria, Ph.D.

Academic Editor

PLOS ONE

Journal Requirements:

"None to declare"

Reviewers' comments:

Reviewer's Responses to Questions

**Comments to the Author**

1. Is the manuscript technically sound, and do the data support the conclusions?

Reviewer #1: Yes

Reviewer #2: Partly

2. Has the statistical analysis been performed appropriately and rigorously? 

Reviewer #1: Yes

Reviewer #2: Yes

3. Have the authors made all data underlying the findings in their manuscript fully available?

Reviewer #1: Yes

Reviewer #2: No

4. Is the manuscript presented in an intelligible fashion and written in standard English?

Reviewer #1: Yes

Reviewer #2: Yes

5. Review Comments to the Author

Reviewer #1: This paper sets out to compare levels and trends over time in alcohol-attributable mortality (AAM) in Finland and Italy using 3 alternative approaches. The authors demonstrate that the choice of approach has a substantial impact on both levels and trends in AAM and propose a new approach which they argue is an improvement on the widely-used Underlying Cause Of Death (UCOD) approach.

Overall this is an interesting and worthwhile thing to do, and the paper is generally clear and well-written, however I have some significant concerns about the conceptual basis of the paper.

Fundamentally the three approaches examined in this paper: UCOD, MCOD and PAF-based are estimating different underlying concepts. UCOD is the minimum set of deaths which we can be all but certain are caused by alcohol as the underlying cause is one which could only be caused by alcohol. MCOD captures all UCOD deaths, plus some deaths which will be certainly attributable to alcohol, but which were not coded as such due to stigma, or clinical error, plus some further deaths where causal attribution to alcohol is more debatable. Finally, the PAF-based approach uses epidemiological evidence combined with data on alcohol consumption to estimate the overall number of deaths from all causes, including those where alcohol is but one of many risk factors. Thus UCOD is a subset of deaths from wholly alcohol-attributable conditions, MCOD is closer to all deaths from wholly alcohol-attributable conditions, but also some deaths that are not directly caused by alcohol and PAF-based is an estimate of all deaths from both wholly and partially alcohol-attributable conditions. The relationship between these three measures will depend on various factors:

• The quality of mortality records, including the prevalence of autopsies

• The level of stigma associated with AAM and the extent to which this might influence clinical coding

• Guidelines and standard practice for recording contributory causes on a death certificate

• Patterns of alcohol consumption (as wholly alcohol-attributable deaths are very strongly associated with heavy drinking, whereas deaths from partially alcohol-attributable causes can occur, albeit at low rates, in people drinking at lower levels – so a population of entirely moderate drinkers will have a different UCOD:PAF ratio than a population with a large number of dependent drinkers)

• The PAFs used, the assumptions inherent in their calculation and whether they vary over time or between population subgroups

The authors touch on some of these issues to some extent, but the conceptual differences between the three measures aren’t well set out in the paper and so it is hard in places to follow the theoretical logic behind the proposed new method. If this logic is not well set out then there is a risk that readers are left with the impression that there is no solid theoretical basis for the new approach, it is simply a mathematical combination of available data that gives a plausible result. Further, the necessary assumptions that must hold in order for the approach, which is based only on Finnish data, to be meaningful for other countries, are not well elaborated.

One other, more minor, point is that some aspects of the analysis of inequalities are underdeveloped. It is almost impossible to discern the trends in inequalities from the graphs and the data is not presented elsewhere. The discussion brings in mention of SII and RII which are not elsewhere defined or presented. If this is an important component of the paper then please provide more detail in the main results section of the text.

Finally, I would suggest that the graphs could be made clearer and more appealing with the use of a little colour.

Reviewer #2: Dear Authors,

Estimation methods in alcohol-attributable mortality are important to the field. I think it’s a valuable contribution to explore other methodologies above the standard PAF (and, sometimes UCOD and MCOD) methods in Finland and Turin.

I don’t think enough has been done to “choose” the MCOD analysis as the gold standard for which to shoot for in the development of your enhanced method. Both the UCOD and MCOD methods entirely ignore (weight to 0.00) all partially-attributable health conditions, like cancer, IHD, stroke and injury, which together make up the majority of AA deaths.

Any similarity the MCOD rates have to the PAF rates seem to simply be a coincidence and not anything that would suggest that the MCOD method is the most desirable. This is my view, in any regard.

Towards the article, I think the discussion about which method is best needs to be built out and expanded. It may be OK to suggest the UCOD Enhanced method as another/competing method, but I’m not sure you go as far as saying it is the “preferred” method, given the enormous limitation of ignoring completely most of the health conditions that are causally related to alcohol.

There are other theoretical limitations (described in more detail below) with the MCOD. In effect, it gives a selective and differential weighting of 1.00 to the alcohol-specific diagnosis regardless of where it appears on the death record. It is different for each record and this is not particularly rigorous or logical, in terms of a weighting strategy.

Abstract

Results

1. I don’t believe the article has done enough to theoretically describe why the MCOD method 2 would provide the most “realistic” rates towards which to shoot.

Conclusions

2. I don’t think this has been developed enough, throughout the article, to state that the new method is preferable. As in my last three comments below, using either UCOD, MCOD, or UCOD Enhanced entirely ignores all the partially-attributable health conditions that together make up the majority of AA death. It doesn’t seem reasonable to put forward any of these methods as the gold standard towards which we should shoot. (Although I recognize the adjustment factor used for the Enhanced method is based on the PAF method).

Introduction

3. First statement. Prefer just “Alcohol consumption” instead of “excessive.” All use *may* carry some risk and it isn’t defined what you mean by “excessive” at this stage.

4. First paragraph. I would request more information about the “different estimation methods” at this stage, since it is motivating the study.

5. Second paragraph. The first sentence here, where two methods are described – there’s not enough information provided to differentiate between the two methods. This paragraph may benefit from first building out the concepts of wholly-attributable and partially-attributable health conditions, and how they relate to your coming analyses.

6. Line 49 – using CCOD variables will definition result in “higher” estimates. But I am not certain they would be “more realistic”. Since the CCOD (what you later call MCOD) analysis weights all diagnosis codes as 1.00 and includes all records as being completely caused by alcohol, it’s hard to know if this is more reliable, or not.

7. End of paragraph two: it would be better if reference 11 applied to more than just the single (small in magnitude) condition of alcoholic cardiomyopathy. But #4 is a great reference for this. Also, here it may be good to state that sometimes the data needed to estimate your own country-specific AAFs (which isn’t particularly difficult if the data is present) is not always available in the country/region in question.

Methods

8. It is a bit odd to have this statement about the Supplementary data and methods at the beginning of the methods section. I would put it where you feel you need to provide more information.

9. Not sure Finland and Italy qualify, in global context, as “vastly different drinking cultures.” Taken in context, they are really fairly similar. Possibly within Europe, they could be considered somewhat different. But Europe is not near representative of global alcohol use. So, some context is needed for this statement.

10. Suggest reorganinzing the subheadings into the following order:

- Data sources, Study population, Approach and existing estimation methods, Novel estimation method.

- It would be better, I believe, to merge the Approach and existing methods description into one.

11. For the smoothing approach. It isn’t clear to me if this is just to re-apportion the 1% of deaths with an unspecific cause or if it’s more general?

12. Can you better describe what the educational levels relate to?

13. Important. For the SES education information, it isn’t clear (in Finland) if this is individually collected (for the census) and then linked to death data. Or if it is area-based (from the census) and attached to individuals via postcode/other method. This is quite important for the study design.

14. Last paragraph of “Data”: these aren’t age groups. It would be helpful to write (30-34, 35—39, and so on)

15. I’m not certain about backwards linear extrapolation of the AAFs. Since the alcohol supply (alcohol per capita) is not included in the estimation, this seems like a bit of a dart throw. Is there a reference that has used this technique before. 1972-1989 (18 years) is a long period to extrapolate!

16. For the UCOD and MCOD methods, it would help to define a term that relates to all the wholly-attributable health condition codes. For examples, sometimes these are called “alcohol-specific” health conditions.

17. In the UCOD paragraph, I really don’t know what this statement could mean “However, we excluded causes of death that were not specific to the age group we studied.” What could you mean here – maybe infant codes, I am not sure.

18. MCOD approach. Should write (100%) alcohol-attributable – i.e. add the 100% or wholly word.

19. Does Finland only have three CCOD variables? If not, why do you only study the first three CCOD variables and not the entire record?

20. For this MCOD approach, it would help to have a statement similar to this “regardless of the position of the alcohol-specific (or whatever term you choose) cause of death ICD10 code, this entire death is categorized as alcohol-attributable. This is analogous to weighting each COD position as 1.00, regardless of where the alcohol-specific code appears.

21. PAF method. Importantly, it isn’t the death that is partially or wholly-attributable, it’s the entire health condition (disease). For example, a breast cancer death is either caused by alcohol (or not), there is no way to know of course it was or not. But we cannot say “8% of this breast cancer death was caused by alcohol.” It is either entirely caused by alcohol or not. We can say that, of 1000 breast cancer deaths, we estimate that about 80 of them occurred because of alcohol use.

22. An example of this difference is in the first sentece of PAF method. We should write “….combine deaths accruing to health conditions which are wholly- and partially-attributable to alcohol use”, instead of what is written about a single death.

23. Also for the PAF method, it isn’t clear for the partially-attributable conditions if these are enumerated only if the code is the UCOD, or if it is also enumerated if the code is anywhere on the abstract (i.e. including MCOD).

24. For negative AAFs – what did you replace them with? Was it simply that you set to 0.00? I would say this is debatable, but it might be OK to truncate them.

Results

25. Figure 1 – typically the y-axes would have the same range in a figure like this. The way you have it may seem to overstate the ASMR in females (since range is 200, instead of 800). Probably, this was to allow to see better the three methods. You could try and come up with a way to let readers know you’ve chosen a different range for females, as compared to males.

26. Figure 2a and 2b. I would not use a line graph for these age group comparisons, I would reserve the line graphs for temporal analyses. It’s quite confusing this way. I would probably makes these two figures into bar graphs for each of the age groups. Also, it cannot be said which is Finland and which is Turin from looking at the figure.

27. Paragraph one – statement “to levels that seem unrealistic.” I would save this interpretation for the discussion.

28. New method. I believe this should be in the “Methods” section, although I understand you wanted to put it after the motivating figures.

29. New method. There really needs to be a formula here, and more information about what the method is doing. For example, it would help to have a table of the “average of these yearly ratios by country and sex over the 1990-2017 period.” This table should be in the article proper (and you don’t even have it in the supplemental), since it’s so critical to the new method. And it would help us as readers understand the ratio-ing work that you’ve done.

30. I think I’m correct that the “correction factor” is (generally) PAF/UCOD for each country/sex, but I am not totally certain that is what was done.

31. In “Development of a novel….” section. Again, I would save the “unrealistic” statement about the PAF method for the discussion and interpretation. Right now, they are simply the findings.

Discussion

Summary of results

32. I’m not certain what “trends in AAM are the same for the two COD-based methods in Finland” is referring to? What do you mean by the trends are the same? There is no metric of this (I don’t think), so do you mean a visual analysis seems to indicate that the MCOD method 2 is similar, but proportionally higher than the UCOD method 1. Or similar?

33. What do you mean by the trends being “less favourable.” I am not sure I follow that section.

34. In this paragraph, it would help to have a clearer description of the new “UCOD enhanced” method. Here, why are the findings from your PAF method “unrealistic” – here’s where a few references are needed as to why those findings are too high for your liking.

35. Last sentence of this subsection – this should be moved to the “appraisal of the new method” section, as you’re getting a bit ahead of the conclusions.

Interpretation of the findings

36. Here again suggest modering the discussion of the differences between Finland and Italy in terms of drinking cultures, since they are quite similar in global perspective. For example, what are the APC levels in each, and at what global percentile is each country? They are probably within 10% (or less) or each other, though I may be wrong.

37. “…our finding that AAM trends are generally more favourable ….” I really don’t know what this could mean.

38. The PAF method is still a (type of) COD method – it just layers another method on top of the COD method, which is still the basis. I think this is lost through the article and would confuse a reader not familiar with all three methods used. E.g. line 293.

39. Line 296, again here I would prefer a broader reference than Manthey et al (this is a good article but only about cardiomyopathy)

40. Around line 300, with the discussion of the PAF / CVD / declining in Europe. It would useful to add a discussion of the fact that, if what you write is true, then one main the reason for the PAF method failing is because the RR functions used are (1) international (and not country-specific, or Europe-specific), and (2) either not age-group specific, or only a few age groups. That would be a suggestion for future work that would help to “fix” the PAF method.

41. At line 307, I’m quite surprised to now find an entirely new analysis (RII and SII) being discussed, which was not presented in the results section, or any of the figures (or tables, if some are added)! Surely, this should be in the results, with accompanying methods in the methods section.

42. In general, there may be too many results to present them all and it could benefit the article to focus on what you deem the most important findings.

43. Limitations are a bit sprinkled throughout the three subsections, it may help to bring them all together under a “strengths and limtations” subsection.

44. Around line 390, Again, it is stated these “overestimations” in the PAF-approach” but this hasn’t been motivated to the strength needed in the rest of the article.

Overall conclusion

45. Hmmm. I’m not sure that you’ve developed the argument that the MCOD method is the most accurate, to use it as the gold standard to shoot for when developing the UCOD Enhanced method for Italy. Recall what the MCOD method is going – it is entirely ignoring all causes of death that are partially-attributable to alcohol (cancer, IHD, stroke, injury, and many more). Together, these make up the vast majority of AA deaths. It seems difficult to believe that a method that weights all of these deaths as 0.00, even though alcohol is known to be causative for all those conditions, is the gold standard that we should reach for.

46. For the MCOD analysis, again what the method does is not very logical. Any death that has any contributing diagnosis from among the ICD10 codes at line 120 is weight as being 100% caused by alcohol use (equivalently, that the death would not have occurred in the absence of exposure to alcohol use). If the death was really 100% caused by alcohol (i.e. there is no doubt that the death would not have occurred in the absence of exposure) why would the alcohol-specific diagnosis not be coded as the UCOD? It doesn’t make a lot of sense to suggest this.

47. At the least, both of these ideas need to be significantly developed in the article rationale, before it would be reasonable to shoot for the MCOD method as the gold standard. At the most, ignoring health conditions that would together for about 70% of AA mortality would be difficult to overcome in providing a new preferred method.

6. PLOS authors have the option to publish the peer review history of their article (what does this mean?). If published, this will include your full peer review and any attached files.

Reviewer #1: No

Reviewer #2: No

---

## [Author Response · Author response to Decision Letter 0]

24 Sep 2023

We thank the reviewers for taking the time to carefully read our manuscript and provide us with their valuable feedback. The manuscript has been thoroughly revised based on the comments received, and we are convinced this has resulted in a substantially improved and more nuanced article. We describe below how have addressed each individual comment. 

Reviewers' comments:

Reviewer #1

1) Fundamentally the three approaches examined in this paper: UCOD, MCOD and PAF-based are estimating different underlying concepts. UCOD is the minimum set of deaths which we can be all but certain are caused by alcohol as the underlying cause is one which could only be caused by alcohol. MCOD captures all UCOD deaths, plus some deaths which will be certainly attributable to alcohol, but which were not coded as such due to stigma, or clinical error, plus some further deaths where causal attribution to alcohol is more debatable. Finally, the PAF-based approach uses epidemiological evidence combined with data on alcohol consumption to estimate the overall number of deaths from all causes, including those where alcohol is but one of many risk factors. Thus UCOD is a subset of deaths from wholly alcohol-attributable conditions, MCOD is closer to all deaths from wholly alcohol-attributable conditions, but also some deaths that are not directly caused by alcohol and PAF-based is an estimate of all deaths from both wholly and partially alcohol-attributable conditions. The relationship between these three measures will depend on various factors:

• The quality of mortality records, including the prevalence of autopsies

• The level of stigma associated with AAM and the extent to which this might influence clinical coding

• Guidelines and standard practice for recording contributory causes on a death certificate

• Patterns of alcohol consumption (as wholly alcohol-attributable deaths are very strongly associated with heavy drinking, whereas deaths from partially alcohol-attributable causes can occur, albeit at low rates, in people drinking at lower levels – so a population of entirely moderate drinkers will have a different UCOD:PAF ratio than a population with a large number of dependent drinkers)

• The PAFs used, the assumptions inherent in their calculation and whether they vary over time or between population subgroups

The authors touch on some of these issues to some extent, but the conceptual differences between the three measures aren’t well set out in the paper and so it is hard in places to follow the theoretical logic behind the proposed new method. If this logic is not well set out then there is a risk that readers are left with the impression that there is no solid theoretical basis for the new approach, it is simply a mathematical combination of available data that gives a plausible result. Further, the necessary assumptions that must hold in order for the approach, which is based only on Finnish data, to be meaningful for other countries, are not well elaborated.

We have made substantial changes in how the existing methods and the new UCOD-Enh. method are introduced and discussed in this paper. First, we have more explicitly introduced the more conceptual differences between the three existing methods that are compared from the introduction onwards (P4L39-57). Second, we have improved the text that includes the theoretical logic of the proposed new method (P10-12). In the discussion, we have addressed the assumptions that must hold in order for the new approach to be meaningful (P20-22).

2) One other, more minor, point is that some aspects of the analysis of inequalities are underdeveloped. It is almost impossible to discern the trends in inequalities from the graphs and the data is not presented elsewhere. The discussion brings in mention of SII and RII which are not elsewhere defined or presented. If this is an important component of the paper then please provide more detail in the main results section of the text.

We have attributed a larger role to the analysis of inequalities following the reviewer’s comment. Indeed, as we demonstrate that the choice of estimation method impacts overall trends and those by educational level, their impact on educational inequalities should also be illustrated more clearly. Details regarding our analysis on inequalities were added to the methods (P10L180-184) and results sections (P16L325-333), we have revised the inequalities graph and have now divided it into two graphs: one for the relative inequality index (Fig 5) and one for the slope index of inequality (Fig 6), which are both included in the main manuscript. We have furthermore added both observed and smoothed values in both graphs to enhance the visibility of the trends. 

3) Finally, I would suggest that the graphs could be made clearer and more appealing with the use of a little colour.

Based on this comment, we have critically reappraised the graphs and their readability. the journal submission guidelines (i.e. RGB (8bit/channel) or grayscale only) and to improve accessibility of the article figures for those suffering from colour blindness, we have decided to keep the grayscale figures as they are.

Reviewer #2

Abstract

Results

1) I don’t believe the article has done enough to theoretically describe why the MCOD method 2 would provide the most “realistic” rates towards which to shoot.

We agree with the reviewer’s overall remark that, in the manuscript as submitted, the MCOD may have been overly referenced as a go-to method for the estimation of alcohol-attributable mortality while it is not without limitation and there is, indeed, no perfect method to estimate alcohol-attributable mortality. We changed our manuscript in line with this remark. 

We would, however, like to note why we find the MCOD method less flawed for the estimation of AAM by educational attainment than the UCOD or PAF-methods:

1) Compared to the UCOD method, the MCOD does provide some indication of mortality due to partly alcohol-attributable conditions by considering the appearance of alcohol-specific causes in other places on a death certificate. Literature from Northern-Ireland and Finland demonstrates that considering contributory causes of death in addition to only the underlying one increases alcohol attributable death counts with about 60% on average, whereby particularly external causes (e.g. self-harm, road traffic accidents, other accidents), cardiovascular conditions, and –in some cases- respiratory conditions are added that would otherwise not be included (1, 2). (view also comments 2, 45 and 47)

2) Compared to the PAF-method, the MCOD fully relies on individual, and therefore education-specific, information from death certificates, whereas the epidemiological and behavioural (i.e. alcohol consumption) information used in the PAF-method is generally not available by educational level. 

Earlier publications both on AAM estimation in general, as well as by educational attainment, have expressed that despite the MCOD’s limitations, it is overall likely to be more accurate and preferred when the required data for estimation (i.e. contributory cause of death information) is available (3).

We have clarified this in our manuscript. 

Conclusions

2) I don’t think this has been developed enough, throughout the article, to state that the new method is preferable. As in my last three comments below, using either UCOD, MCOD, or UCOD Enhanced entirely ignores all the partially-attributable health conditions that together make up the majority of AA death. It doesn’t seem reasonable to put forward any of these methods as the gold standard towards which we should shoot. (Although I recognize the adjustment factor used for the Enhanced method is based on the PAF method).

In our manuscript, we have removed any suggestion that there is a gold standard method for alcohol-attributable mortality and have more carefully expressed when – and compared to which method (i.e. UCOD and PAF) - the newly developed method likely presents a better alternative. We, however, do not believe that the MCOD- or UCOD-Enh. method entirely exclude partly AAM. Regarding MCOD, we kindly refer back to our reply to comment 1 for more details. Regarding UCOD-Enh., the ratio provides a proxy for partly AAM. We have expanded on the theoretical underpinnings of the UCOD-Enh. method that clarify this in the methods section (‘Development of an alternative estimation method’). 

Introduction

3) First statement. Prefer just “Alcohol consumption” instead of “excessive.” All use *may* carry some risk and it isn’t defined what you mean by “excessive” at this stage.

We applied this suggestion.

4) First paragraph. I would request more information about the “different estimation methods” at this stage, since it is motivating the study.

We have added more information about these estimation methods, as well as their different levels and age patterns by estimation method to provide more context (P4L35-P5L63).

5) Second paragraph. The first sentence here, where two methods are described – there’s not enough information provided to differentiate between the two methods. This paragraph may benefit from first building out the concepts of wholly-attributable and partially-attributable health conditions, and how they relate to your coming analyses.

We have rewritten the introduction (P4-6), and introduced the concepts of wholly and partly alcohol-attributable conditions and mortality, as well as how different estimation approaches try to estimate these concepts, between P4L39-57. We believe the introduction indeed benefits from the earlier introduction of these concepts and thank the reviewer for this observation.

6) Line 49 – using CCOD variables will definition result in “higher” estimates. But I am not certain they would be “more realistic”. Since the CCOD (what you later call MCOD) analysis weights all diagnosis codes as 1.00 and includes all records as being completely caused by alcohol, it’s hard to know if this is more reliable, or not.

We removed the part “and hence more reliable” from this sentence, and have refrained from using this term or the term “realistic” from here on out (including in the supporting information files).

7) End of paragraph two: it would be better if reference 11 applied to more than just the single (small in magnitude) condition of alcoholic cardiomyopathy. But #4 is a great reference for this. Also, here it may be good to state that sometimes the data needed to estimate your own country-specific AAFs (which isn’t particularly difficult if the data is present) is not always available in the country/region in question.

We have removed reference 11 from this paragraph, as well as in other places where the limitations of PAFs at older ages are discussed (i.e. P5L76, and supporting information 2 throughout). The following sentence was added upon the reviewer’s suggestion:

“Furthermore, the data needed to estimate PAFs that are both education- and country-specific (i.e. alcohol consumption by age, sex, and educational attainment; relative mortality risks at different levels of alcohol consumption) is rarely available in the country or region of study.” (P6L79-82).

Methods

8) It is a bit odd to have this statement about the Supplementary data and methods at the beginning of the methods section. I would put it where you feel you need to provide more information.

This has been removed, we now refer to the supporting information file where necessary.

9) Not sure Finland and Italy qualify, in global context, as “vastly different drinking cultures.” Taken in context, they are really fairly similar. Possibly within Europe, they could be considered somewhat different. But Europe is not near representative of global alcohol use. So, some context is needed for this statement.

Although from a global perspective differences in alcohol consumption between the two countries might appear small (8.4 litres of pure alcohol consumption in litres in Finland for ages 15 and older, 7.4 in Italy in 2017)(5), within Europe and especially from a more historical perspective they exhibit vastly different drinking cultures. Consumption has strongly declined in Italy (19.5 litres in 1972), but moderately increased in Finland (6.8 litres in 1972) over the study period, which would affect our results on trends in AAM (4). In addition, heavy episodic drinking is more common in Finland, with intoxication generally being considered an important part of alcohol culture in Finland (5-7) , while in Italy alcohol is more traditionally consumed during meals (8). We believe each of these characteristics of ‘drinking culture’ may contribute to differences in alcohol-attributable mortality between both settings, given the alcohol-attributable conditions each population would suffer from would differ (e.g. more acute causes in Finland compared to more chronic ones in Italy), as would the subsequent age patterns and inequalities.

We have added this contextual information to the ‘study population’ paragraph in the Methods-section (P7L122-P8L134).

10) Suggest reorganizing the subheadings into the following order:

- Data sources, Study population, Approach and existing estimation methods, Novel estimation method.

- It would be better, I believe, to merge the Approach and existing methods description into one.

We appreciate the reviewer’s suggestion and have changed the organisation of the methods section accordingly.

11) For the smoothing approach. It isn’t clear to me if this is just to re-apportion the 1% of deaths with an unspecific cause or if it’s more general?

The smoothing approach was only applied to the Turin longitudinal study (TLS) data, and aims to deal with the heavy fluctuations in data due to small cells. The re-apportion of the 1% deaths with an unspecific cause after smoothing further aims to optimise the information available for analysis from the TLS, and was performed after smoothing.

12) Can you better describe what the educational levels relate to?

We have added more information regarding the educational levels:

“In both countries the study population is subdivided by SES according to individually collected census information on the highest level of completed education, whereby we consider International Standard Classification of Education 1997 (ISCED) ‘Lower-’ (ISCED 0-2; pre-primary to lower secondary education), ‘Middle-’ (ISCED 3-4; upper secondary to post-secondary non-tertiary education), and ‘Higher-’educated (ISCED 5-6; tertiary education) categories (9).” (P7L104-108)

13) Important. For the SES education information, it isn’t clear (in Finland) if this is individually collected (for the census) and then linked to death data. Or if it is area-based (from the census) and attached to individuals via postcode/other method. This is quite important for the study design.

The educational information for Finland was, indeed, also based on individual census information. We have made this more explicit by adding “individually collected census information” (see also point 12).

14) Last paragraph of “Data”: these aren’t age groups. It would be helpful to write (30-34, 35—39, and so on)

We have changed this to: “… and five year age group (30-34, 35-39, 40-44, and so on, up to 95+).” (P7L116-117).

15) I’m not certain about backwards linear extrapolation of the AAFs. Since the alcohol supply (alcohol per capita) is not included in the estimation, this seems like a bit of a dart throw. Is there a reference that has used this technique before. 1972-1989 (18 years) is a long period to extrapolate!

The backwards linear extrapolation of PAFs was primarily based on the observed linearity in the age-, sex-, and country-specific trends in PAFs from the GBD. We indeed did not dispose of more specific literature or data to inform the extrapolation of the alcohol attributable fractions used in the PAF-method. Given the reviewer’s concern, and because the AAM levels for 1972-1989 are not strictly necessary to fulfil the aims of this paper, we decided to remove all findings relying on these extrapolated PAFs, limiting these levels, trends, age patterns, and educational inequalities in the PAF-method between 1990 and 2017. This did not affect the estimation of AAM based on our alternative method because, given the uncertainty regarding the extrapolated PAFs, we calculated the ratios for the UCOD-Enh. method from 1990 onwards.

16) For the UCOD and MCOD methods, it would help to define a term that relates to all the wholly-attributable health condition codes. For examples, sometimes these are called “alcohol-specific” health conditions.

We have applied the term “alcohol-specific” (P8L148, P9L159).

17) In the UCOD paragraph, I really don’t know what this statement could mean “However, we excluded causes of death that were not specific to the age group we studied.” What could you mean here – maybe infant codes, I am not sure.

This, indeed, refers to infant codes. We have added the following to make this more clear: “However, we excluded causes of death that were not relevant for those aged 30 and older (e.g. P04.3 Foetus and new-born affected by maternal use of alcohol, Q86.0 Foetal alcohol syndrome)”. (P8L149-P9L150)

18) MCOD approach. Should write (100%) alcohol-attributable – i.e. add the 100% or wholly word.

We have added “wholly” (P9L157). 

19) Does Finland only have three CCOD variables? If not, why do you only study the first three CCOD variables and not the entire record?

Only the first three contributory causes are recorded for statistical purposes, as 4 or more contributory causes of death are exceedingly rare. In other Finnish research on alcohol-attributable mortality by SES over time, this same approach was adopted (10, 11). The first contributory cause that contains an alcohol-specific condition is furthermore selected when multiple relevant conditions appear for subsequent contributory causes. We argue that the earlier the alcohol-specific condition appears on a death certificate, the larger its importance for alcohol-attributable mortality. 

20) For this MCOD approach, it would help to have a statement similar to this “regardless of the position of the alcohol-specific (or whatever term you choose) cause of death ICD10 code, this entire death is categorized as alcohol-attributable. This is analogous to weighting each COD position as 1.00, regardless of where the alcohol-specific code appears.

The following was added: “Regardless of where on the death certificate the alcohol-specific cause appears, the death is included in its entirety.”

21) PAF method. Importantly, it isn’t the death that is partially or wholly-attributable, it’s the entire health condition (disease). For example, a breast cancer death is either caused by alcohol (or not), there is no way to know of course it was or not. But we cannot say “8% of this breast cancer death was caused by alcohol.” It is either entirely caused by alcohol or not. We can say that, of 1000 breast cancer deaths, we estimate that about 80 of them occurred because of alcohol use. 

We have rephrased this section on the PAF-method and believe it better captures the condition- rather than single death-focused nature of this method (P9L162-167).

22) An example of this difference is in the first sentence of PAF method. We should write “….combine deaths accruing to health conditions which are wholly- and partially-attributable to alcohol use”, instead of what is written about a single death.

We have altered this sentence as proposed..

23) Also for the PAF method, it isn’t clear for the partially-attributable conditions if these are enumerated only if the code is the UCOD, or if it is also enumerated if the code is anywhere on the abstract (i.e. including MCOD). 

We have changed the text to “PAF-estimates are thereby obtained by multiplying death counts from an elaborate list of alcohol-related conditions that appear as the underlying cause on death certificates, with their respective PAFs from the GBD (12).”, to specify that the PAF method only considers conditions appearing as underlying causes. 

24) For negative AAFs – what did you replace them with? Was it simply that you set to 0.00? I would say this is debatable, but it might be OK to truncate them.

We indeed set these AAFs to 0.00 (this has now been specified in the manuscript aside from the existing mention in Supporting information 1: Data & Methods). We did so to remove any assumptions of so-called ‘cardioprotective effects’ of alcohol consumption from our comparison, as was done by Trias-Llimos et al 2018, given the strong debate about this phenomenon in the literature (13). We referred to this ongoing debate on P9L171-172. Given we found no sources that justified another type of truncation of AAFs to accommodate for the existing academic debate about cardioprotective effects of alcohol, we opted to exclude them from this paper altogether.

Results

25) Figure 1 – typically the y-axes would have the same range in a figure like this. The way you have it may seem to overstate the ASMR in females (since range is 200, instead of 800). Probably, this was to allow to see better the three methods. You could try and come up with a way to let readers know you’ve chosen a different range for females, as compared to males.

We indeed allowed for the Y-axis range to differ to ensure visibility of our comparisons. In order to clarify this, we have now added “The Y-axis scale differs by sex to improve visibility of the results” to the figure captions. For figures 3 and 4 we added “The Y-axis scale differs by sex and country to improve visibility of the results”.

26) Figure 2a and 2b. I would not use a line graph for these age group comparisons, I would reserve the line graphs for temporal analyses. It’s quite confusing this way. I would probably makes these two figures into bar graphs for each of the age groups. Also, it cannot be said which is Finland and which is Turin from looking at the figure.

We have changed these figures to bar graphs and have included the country name on top.

27) Paragraph one – statement “to levels that seem unrealistic.” I would save this interpretation for the discussion.

This part of the sentence was removed.

28) New method. I believe this should be in the “Methods” section, although I understand you wanted to put it after the motivating figures.

We have moved the parts of the development of the new method from the results to the methods section, and focused the results on the actual comparison of the new method to existing ones. 

29) New method. There really needs to be a formula here, and more information about what the method is doing. For example, it would help to have a table of the “average of these yearly ratios by country and sex over the 1990-2017 period.” This table should be in the article proper (and you don’t even have it in the supplemental), since it’s so critical to the new method. And it would help us as readers understand the ratio-ing work that you’ve done.

We have now added the average ratios by country and sex in the manuscript methods section (P12L225-226), and have included a table with the yearly and average ratios between 1990 and 2017 in supporting information file 2 (Table S2.1). Two equations have also been added to both the manuscript and the supporting information, one to illustrate the calculation of the ratio and one for the application of the average ratio to the UCOD death counts.

30) I think I’m correct that the “correction factor” is (generally) PAF/UCOD for each country/sex, but I am not totally certain that is what was done.

This paragraph was moved to the methods section, and we have now added “The level-adjustment consists of a ratio-based approach, whereby the ratio summarises the relationship between UCOD (wholly AAM) and PAF (wholly and partly AAM) between 1990 and 2017 (age-standardised death rates [ASD] truncated for ages 30-64) [fomula 2]. The average ratio thereby amounts to 1.98 among Finnish males, 1.86 for Finnish females, and 3.43 and 3.33 for Italian males and females, respectively.” to be more specific (see also comment 29).

31) In “Development of a novel….” section. Again, I would save the “unrealistic” statement about the PAF method for the discussion and interpretation. Right now, they are simply the findings.

The word “unrealistic” was removed, this was rephrased to “In order to compare trends and levels in AAM in future comparative studies, we propose a new method that deals with the under-estimation of AAM in the UCOD method and avoids the use of suboptimal data to estimate PAFs beyond age 65 in the PAF-method.”.

Discussion

Summary of results

32) I’m not certain what “trends in AAM are the same for the two COD-based methods in Finland” is referring to? What do you mean by the trends are the same? There is no metric of this (I don’t think), so do you mean a visual analysis seems to indicate that the MCOD method 2 is similar, but proportionally higher than the UCOD method 1. Or similar?

This indeed pertains to a visual analysis of the trends over time, whereby the actual curve for UCOD and MCOD are highly similar, albeit at different levels of AAM. We slightly rewrote this sentence as “first, visual analysis of trends yields highly similar curves when only COD data is used, both in the general population and by educational level, albeit at different levels.”

33) What do you mean by the trends being “less favourable.” I am not sure I follow that section.

We have more explicitly described which findings we refer to when trends are said to be ‘less favourable’ by rewriting this part as follows: “UCOD and MCOD thereby showed mainly increasing trends in AAM according to UCOD and MCOD compared to a generally declining trend according to the PAF-method.” (P17L355-356)

34) In this paragraph, it would help to have a clearer description of the new “UCOD enhanced” method. Here, why are the findings from your PAF method “unrealistic” – here’s where a few references are needed as to why those findings are too high for your liking.

We no longer refer to limitations of the PAF method in this part of the discussion (summary of results), and also no longer refer to results as being unrealistic or too high. Rather, from P19L385 to L393 we point to general issues with PAF-based results.

35) Last sentence of this subsection – this should be moved to the “appraisal of the new method” section, as you’re getting a bit ahead of the conclusions.

We have removed this sentence from this subsection and now discuss when UCOD-Enh. provides a good alternative in the ‘appraisal of the new method section’, as advised by the reviewer, as well as in the conclusion.

Interpretation of the findings

36) Here again suggest modering the discussion of the differences between Finland and Italy in terms of drinking cultures, since they are quite similar in global perspective. For example, what are the APC levels in each, and at what global percentile is each country? They are probably within 10% (or less) or each other, though I may be wrong.

See comment 9

37) “…our finding that AAM trends are generally more favourable ….” I really don’t know what this could mean.

We have specified what this means: “Our finding that AAM trends are generally more favourable using a PAF-based approach compared to a COD-approach (due to general declines in Finland for males and steeper declines in Italy than using another method) is most likely partly due to the inclusion of a range of cardiovascular diseases (CVDs) (e.g. ischaemic heart disease, stroke, hypertension) in PAF methods that are not typically included in COD-based methods […]” (P18L372-376).

38) The PAF method is still a (type of) COD method – it just layers another method on top of the COD method, which is still the basis. I think this is lost through the article and would confuse a reader not familiar with all three methods used. E.g. line 293.

Throughout the article, we have removed the use of the words ‘COD-approaches’ and ‘PAF-approaches’. We now more often distinguish between approaches that measure only wholly AAM (UCOD) or wholly and partly AAM (MCOD, PAF, UCOD-Enh.). Whenever UCOD and MCOD are not explicitly named, but discussed with an overall term, we refer to them as ‘solely COD-based’ or ‘relying on COD information from death certificates only’. 

39) Line 296, again here I would prefer a broader reference than Manthey et al (this is a good article but only about cardiomyopathy) (see also 7)

Similar to comment 7, we have removed this reference and kept the reference to Trias-llímos et al. 2018. 

40) Around line 300, with the discussion of the PAF / CVD / declining in Europe. It would useful to add a discussion of the fact that, if what you write is true, then one main the reason for the PAF method failing is because the RR functions used are (1) international (and not country-specific, or Europe-specific), and (2) either not age-group specific, or only a few age groups. That would be a suggestion for future work that would help to “fix” the PAF method.

We have added the following statement (P19L385-393): “Importantly, the trends by educational attainment we demonstrated for the PAF-method are generally less trustworthy. The data required to come by a completely country- and education-specific PAF-estimate are difficult to come by, namely alcohol consumption data that is specific to educational attainment, and education- and country-specific relative mortality risks by different levels of alcohol consumption compared to abstainers. Future research relying on PAF-methods to estimate AAM by educational level would benefit from overall improvements in the specificity of PAFs by age group and country (particularly in the relative mortality risks used) in order to avoid bias in the role attributed to alcohol in mortality trends over time, but more crucially also from the development of education-specific fractions.”. 

41) At line 307, I’m quite surprised to now find an entirely new analysis (RII and SII) being discussed, which was not presented in the results section, or any of the figures (or tables, if some are added)! Surely, this should be in the results, with accompanying methods in the methods section.

We have attributed a larger role to the analysis of inequalities following the reviewer’s comment. Indeed, as we demonstrate that the choice of estimation method impacts overall trends and those by educational level, their impact on educational inequalities should also be illustrated more clearly. Details regarding our analysis on inequalities were added to the methods (P10L180-184) and results sections (P16L325-333), we have revised the inequalities graph and have now divided it into two graphs: one for the relative inequality index (Fig 5) and one for the slope index of inequality (Fig 6), which are both included in the main manuscript. We have furthermore added both observed and smoothed values in both graphs to enhance the visibility of the trends.

42) In general, there may be too many results to present them all and it could benefit the article to focus on what you deem the most important findings.

We critically reappraised the contents of the results section – also related to the previous remark. We concluded that the analysis of inequalities is key for the paper (see previous reply), and therefore included it as part of the results. In our view all results and figures currently presented are equally important for our paper, so we did not make any further changes to the results section. We did structure the results with a subheading for each research objective to make them easier to identify in this section, and shortened the ‘summary of results’ subsection in the discussion.

43) Limitations are a bit sprinkled throughout the three subsections, it may help to bring them all together under a “strengths and limitations” subsection.

We have applied this suggestion and have now added a limitations section on P22-23.

44) Around line 390, Again, it is stated these “overestimations” in the PAF-approach” but this hasn’t been motivated to the strength needed in the rest of the article.

We no longer refer to overestimations in PAF, but to limitations to PAFs at older ages in general and the lack of education-specificity of PAFs.

Overall conclusion

45) Hmmm. I’m not sure that you’ve developed the argument that the MCOD method is the most accurate, to use it as the gold standard to shoot for when developing the UCOD Enhanced method for Italy. Recall what the MCOD method is going – it is entirely ignoring all causes of death that are partially-attributable to alcohol (cancer, IHD, stroke, injury, and many more). Together, these make up the vast majority of AA deaths. It seems difficult to believe that a method that weights all of these deaths as 0.00, even though alcohol is known to be causative for all those conditions, is the gold standard that we should reach for. (See also 1)

See our reply to comment 1

46) For the MCOD analysis, again what the method does is not very logical. Any death that has any contributing diagnosis from among the ICD10 codes at line 120 is weight as being 100% caused by alcohol use (equivalently, that the death would not have occurred in the absence of exposure to alcohol use). If the death was really 100% caused by alcohol (i.e. there is no doubt that the death would not have occurred in the absence of exposure) why would the alcohol-specific diagnosis not be coded as the UCOD? It doesn’t make a lot of sense to suggest this.

Contributory causes of death should indeed be understood as antecedent conditions giving rise to the underlying cause or contributing to it, whereby, the underlying cause is unlikely to occur without the contributory one (14). This ’inevitability’ is however difficult to establish retrospectively for behavioural risk factors in particular, despite the extensive coding instructions from WHO, and that could be one reason why the alcohol-specific diagnosis is regularly coded as contributory instead of underlying

Regarding MCOD, it is thereby important to take into account its generally different logic from the PAF-method. Rather than aiming to quantify which portion of all conditions appearing as underlying causes on a certificate is logically attributable to alcohol consumption according to elaborate comparative risk assessment, MCOD tries to capture those instances where non-alcohol-specific conditions are caused by alcohol consumption (i.e. partly alcohol-attributable conditions). Prior research on this has in fact already argued that the association between risk factors and death are not sensitive to the placement of the risk factor (for us alcohol-specific conditions) on the death certificate (15). 

47) At the least, both of these ideas need to be significantly developed in the article rationale, before it would be reasonable to shoot for the MCOD method as the gold standard. At the most, ignoring health conditions that would together for about 70% of AA mortality would be difficult to overcome in providing a new preferred method 

See our reply to comment 1.

References

1. Durkin A, Connolly S, O’Reilly D. Quantifying Alcohol-Related Mortality: Should Alcohol-Related Contributory Causes of Death be Included? Alcohol and Alcoholism. 2010;45(4):374-8.

2. Mäkelä P, Valkonen T, Martelin T. Contribution of deaths related to alcohol use to socioeconomic variation in mortality: register based follow up study. BMJ: the British Medical Journal. 1997;315(7102).

3. Taylor B, Rehm J, Room R, Patra J, Bondy S. Determination of Lifetime Injury Mortality Risk in Canada in 2002 by Drinking Amount per Occasion and Number of Occasions. American Journal of Epidemiology. 2008;168(10):1119-25.

4. WHO. European Health for All database (HFA-DB) data source Copenhagen: World Health Organization Regional Office for Europe; 2022 [Available from: https://gateway.euro.who.int/en/datasets/european-health-for-all-database/.

5. Moskalewicz J, Room R, Thom B. Comparative monitoring of alcohol epidemiology across the EU. Baseline assessment and suggestions for future actions. Synthesis report. Warsaw: PARPA - The State Agency for Prevention of Alcohol Related Problems; 2016.

6. Mäkelä P, Kumpulainen P, Härkönen J, Lintonen T. How Are All Drinking Occasions, Intoxication Occasions, and All Alcohol Consumed Distributed Across Different Drinking Occasion Types? A Typology of Drinking Occasions in Finland. Journal of studies on alcohol and drugs. 2021;82(6):767-75.

7. Mäkelä p, Tigerstedt C, Mustonen H. The Finnish drinking culture: Change and continuity in the past 40 years. Drug Alcohol Rev. 2012;31(7):831-40.

8. Allamani A, Pepe P, Baccini M, Massini G, Voller F. Europe. An Analysis of Changes in the Consumption of Alcoholic Beverages: The Interaction Among Consumption, Related Harms, Contextual Factors and Alcoholic Beverage Control Policies. Substance Use & Misuse. 2014;49(12):1692-715.

9. UNESCO. International standard classification of education-ISCED 1997 UNESCO; 1997. Contract No.: Report.

10. Martikainen P, Mäkelä P, Peltonen R, Myrskyla M. Income differences in life expectancy: the changing contribution of harmful consumption of alcohol and smoking. Epidemiology. 2014;25(2):182-90.

11. Trias-Llimós S, Spijker JJA. Educational differences in alcohol-related mortality and their impact on life expectancy and lifespan variation in Spain (2016–2018): a cross-sectional analysis using multiple causes of death. BMJ Open. 2022;12(e053205).

12. GBD 2016 Risk Factors Collaborators. Global, regional, and national comparative risk assessment of 84 behavioural, environmental and occupational, and metabolic risks or clusters of risks, 1990–2016: a systematic analysis for the Global Burden of Disease Study 2016. The Lancet. 2017;390(10100):1345-422.

13. van de Luitgaarden IAT, van Oort S, Bouman EJ, Schoonmade LJ, Schrieks IC, Grobbee DE, et al. Alcohol consumption in relation to cardiovascular diseases and mortality: a systematic review of Mendelian randomization studies. European Journal of Epidemiology. 2021;37(7):655-69.

14. Eurostat. Causes of death (hlth_cdeath) 2023 [Available from: https://ec.europa.eu/eurostat/cache/metadata/en/hlth_cdeath_sims.htm.

15. Batty GD, Gale CR, Kivimäki M, Bell S. Assessment of Relative Utility of Underlying vs Contributory Causes of Death. JAMA Network Open. 2019;2(7):e198024-e.

---

## [Decision Letter · Decision Letter 1]

29 Nov 2023

The impact of estimation methods for alcohol-attributable mortality on long-term trends for the general population and by educational level in Finland and Italy (Turin)

PONE-D-23-15590R1

Dear Dr. Van Hemelrijck,

We’re pleased to inform you that your manuscript has been judged scientifically suitable for publication and will be formally accepted for publication once it meets all outstanding technical requirements.

Kind regards,

Sina Azadnajafabad, MD, MPH

Academic Editor

PLOS ONE

Additional Editor Comments (optional):

Reviewers' comments:

Reviewer's Responses to Questions

**Comments to the Author**

1. If the authors have adequately addressed your comments raised in a previous round of review and you feel that this manuscript is now acceptable for publication, you may indicate that here to bypass the “Comments to the Author” section, enter your conflict of interest statement in the “Confidential to Editor” section, and submit your "Accept" recommendation.

Reviewer #2: All comments have been addressed

2. Is the manuscript technically sound, and do the data support the conclusions?

Reviewer #2: Yes

3. Has the statistical analysis been performed appropriately and rigorously? 

Reviewer #2: Yes

4. Have the authors made all data underlying the findings in their manuscript fully available?

Reviewer #2: Yes

5. Is the manuscript presented in an intelligible fashion and written in standard English?

Reviewer #2: Yes

6. Review Comments to the Author

Reviewer #2: You have rigourously addressed my proposed comments and changes, and are commended for a strong article!

7. PLOS authors have the option to publish the peer review history of their article (what does this mean?). If published, this will include your full peer review and any attached files.

Reviewer #2: **Yes: **Adam Sherk

---

## [Editor Report · Acceptance letter]

6 Dec 2023

PONE-D-23-15590R1 

The impact of estimation methods for alcohol-attributable mortality on long-term trends for the general population and by educational level in Finland and Italy (Turin) 

Dear Dr. Van Hemelrijck:

I'm pleased to inform you that your manuscript has been deemed suitable for publication in PLOS ONE. Congratulations! Your manuscript is now with our production department. 

Kind regards, 

on behalf of

Dr. Sina Azadnajafabad 

Academic Editor

PLOS ONE